# Efficient Exploration and Discriminative World Model Learning with an Object-Centric Abstraction

**Anthony GX-Chen**
Center for Data Science
New York University
anthony.gx.chen@nyu.edu

**Kenneth Marino**
Google DeepMind
The University of Utah

**Rob Fergus**
Dept. of Computer Science
New York University

## Abstract

In the face of difficult exploration problems in reinforcement learning, we study whether giving an agent an object-centric mapping (describing a set of *items* and their *attributes*) allow for more efficient learning. We found this problem is best solved hierarchically by modelling items at a higher level of state abstraction to pixels, and attribute change at a higher level of temporal abstraction to primitive actions. This abstraction simplifies the transition dynamic by making specific future states easier to predict. We make use of this to propose a fully model-based algorithm that learns a discriminative world model, plans to explore efficiently with only a count-based intrinsic reward, and can subsequently plan to reach any discovered (abstract) states.

We demonstrate the model's ability to (i) efficiently solve single tasks, (ii) transfer zero-shot and few-shot across item types and environments, and (iii) plan across long horizons. Across a suite of 2D crafting and MiniHack environments, we empirically show our model significantly out-performs state-of-the-art low-level methods (without abstraction), as well as performant model-free and model-based methods using the same abstraction. Finally, we show how to learn low level object-perturbing policies via reinforcement learning, and the object mapping itself by supervised learning.

## 1 Introduction

Since the inception of reinforcement learning (RL), the problems of exploration and world model learning have been active, open questions. RL requires an agent to learn both basic motor abilities, and explore long sequences of interactions in the environment. Currently, we have made tremendous progress on problems of low-level control for short-horizon problems. With well defined single tasks, given enough samples (or demonstrations), there are well-developed ways of training low level policies reliably (Schulman et al., 2017; Hafner et al., 2023).

Perhaps a better way to explore complex environments is to treat the high-level exploration problem separately. We increasingly have the ability to get semantically rich abstractions: via representation learning (Locatello et al., 2020), object segmentation (Kirillov et al., 2023), and visual-language models (Alayrac et al., 2022); even in difficult control settings such as robotics (Liu et al., 2024). In addition, with the recent explosion in the field of natural language processing, people are increasingly interested in addressing RL environments on a semantic level (Andreas et al., 2017; Jiang et al., 2019; Chen et al., 2021). Thus, we ask a timely question: *if* the agent has a good ability to perceive objects, what *more* can we do other than tabula rasa learning at the level of pixels and primitive motor actions?

In this work, we focus on exploration and world modelling at a semantic level. We explore a simple yet flexible abstraction: in addition to the agent's raw observations (e.g. pixels), it sees an abstract representation of a set of *items* and their *attributes* (i.e. object states). We find that to navigate these two levels effectively, it is best to construct a proxy decision process (Ab-MDP) with state and temporal abstractions at the level of objects, with a handful of competent object-perturbing low level policies. By construction, dynamics on this semantic level becomes structured, and we

make use of this to design a *discriminative* model-based RL method that learns to predicts if a specific object perturbation will be *successful*. Our method, **M**odel-based **E**xploration of abstracted **A**ttribute **D**ynamics (**MEAD**), is fit with a simple objective to model semantic level transitions, resulting in stable model improvements with no need for auxiliary objectives. It is fully online: the agent plans to explore without an extrinsic reward to learn about the world it inhabits using a count-based objective. The same model can be used at any point to plan to solve downstream tasks when given a goal function, without the need to train policy or value functions.

The main results of this work investigates decision making at the semantic level as defined by the Ab-MDP, assuming access to an object centric map and object perturbing policies. We empirically evaluate our method in a set of 2D crafting games, and MiniHack environments that are known to be very difficult to explore (Henaff et al., 2022). We then show how to *learn* components of the Ab-MDP when it is not given: by reinforcement learning object perturbation policies given an object mapping, and supervised learning of the object centric mapping itself. We leave the unsupervised discovery of the object centric mapping as complementary and future work.

The main contributions of this work are:

- We define the Ab-MDP: a simple human-understandable hierarchical MDP. The structure of this abstraction allows for pre-trained or from-scratch visual encoders to be used. It also allows our method (MEAD) to use a discriminative objective for world-model learning.
- We introduce our model-learning method (MEAD). MEAD uses a stable discriminative objective to model item attribute changes, which is more efficient at learning than generative-based baselines and also allows for efficient inference-time planning.
- We empirically show MEAD extracts an interpretable world model, accurately plans over long horizons, reaches competitive final performance on all levels with greater sample efficiency to strong baselines, and transfers well to new environments.
- We demonstrate how to learn parts of the Ab-MDP when it is not given: by reinforcement learning object-perturbing policies, and supervised learning the object-centric mapping.

## 2 PROBLEM SETTING

We begin by considering a (reward free) "base" Markov Decision Process (MDP) as the tuple $\langle \mathcal{S}, \mathcal{A}, P \rangle$, with state space $\mathcal{S}$, primitive action space $\mathcal{A}$, and transition probability function $P : \mathcal{S} \times \mathcal{A} \times \mathcal{S} \to [0, 1]$ (Puterman, 1994). Below we construct the "abstracted" space.

### 2.1 ABSTRACT STATES AND BEHAVIOURS

We give the agent a deterministic object-centric mapping M: $\mathcal{S} \to \mathcal{X}$ from low level observations to an **abstract state** $X \in \mathcal{X}$. The abstract state is a discrete *set* of items, each having form (item identity, item attribute). Denote the $i$-th item in this (ordered) set as $x^{(i)}$, each $x^{(i)} = (\alpha^{(i)}, \xi^{(i)})$ consists of an item's *identity* $\alpha^{(i)}$ (e.g. "potion"), and *attribute* $\xi^{(i)}$ (e.g. "in inventory"). In practice, each $X$ is represented as a $N \times (d_{\text{iden}} + d_{\text{attr}})$ matrix containing $N$ items (including empty items) each having a $d_{\text{iden}}$ dimensional identity embedding and $d_{\text{attr}}$ dimensional attribute embedding.

Denote a space $\mathcal{B}$ of object perturbation policies which we call **behaviours**. Each behaviour has an associated $\pi_b : \mathcal{S} \to \mathcal{A}$. A behaviour $b = (\alpha^{(i)}, \xi')$ describes a single item id $\alpha^{(i)}$ and a desired *new* attribute. For $N$ objects and $m$ attributes, there are $N \times m$ single item behaviours.[1] Not all behaviours changes $\alpha^{(i)}$ to have $\xi'$ (due to the change being impossible from current $X$, or having a bad $\pi_b$). A behaviour is **competent** if executing behaviour policy $\pi_b$ from state $S$, M$(S) = X$, result in arriving at state $S'$, M$(S') = X'$, $(\alpha^{(i)}, \xi') \in X'$, within $k$ steps with high probability. See Appendix B.1 for more details. Executing competent behaviours lead to predictable changes.

For the main experiments we assume access to (learned or given) abstract states and behaviours and build on top. We address learning both in Section 4.3.

### 2.2 ABSTRACTED ITEM-ATTRIBUTE MDP

We now define an Abstracted Item-Attribute MDP using the abstract state $\mathcal{X}$ and behaviour $\mathcal{B}$ spaces, which we refer to in brief as **Ab-MDP**. The abstract mapping M provides an *state abstraction*. We

---

[1]Here we define *single item* behaviours. Behaviours can also be multi items, see Appendix B.1.5.

further define the Ab-MDP to have a coarser temporal resolution than the low level MDP: an *abstract transition* only occurs when an abstract state changes, or if the state stays the same but $k$ low level steps elapses (see Definition B.1 for more details).

Specifically, the **Ab-MDP** is the tuple $\langle \mathcal{X}, \mathcal{B}, \mathbf{T} \rangle$, with abstract states $\mathcal{X}$, abstract behaviours $\mathcal{B}$, and transition probability function $\mathbf{T} : \mathcal{X} \times \mathcal{B} \times \mathcal{X} \rightarrow [0, 1]$. $\mathbf{T}(X'|X, b)$ describes the probability of being in next abstract state $X'$, when starting from $X$, executing behaviour $b$, and waiting until an abstract transition occurs. Figure 1 provides an example of an abstract transition and associated low level steps. The Ab-MDP can be interpreted via the Options Framework (Sutton et al., 1999; Precup, 2000), which we discuss in Appendix B.1.1.

There is an intuitive relationship between the *competence* of a behaviour and the transition probability function $\mathbf{T}$. For instance, executing an incompetent behaviour $b$ from abstract state $X$ will likely lead to the next abstract state being itself, $X_{t+1} = X_t$, or some hard-to-predict distribution over next states. Executing a competent behaviour $b$ lead to a predictable next state that contains the proposed attribute change with high probability.

In any case, we can treat the Ab-MDP as a regular MDP use standard RL algorithms to solve it. This can be viewed as solving a proxy problem of the low level base MDP. **For the remainder of this paper, all methods are trained *exclusively* within the Ab-MDP** unless stated otherwise.

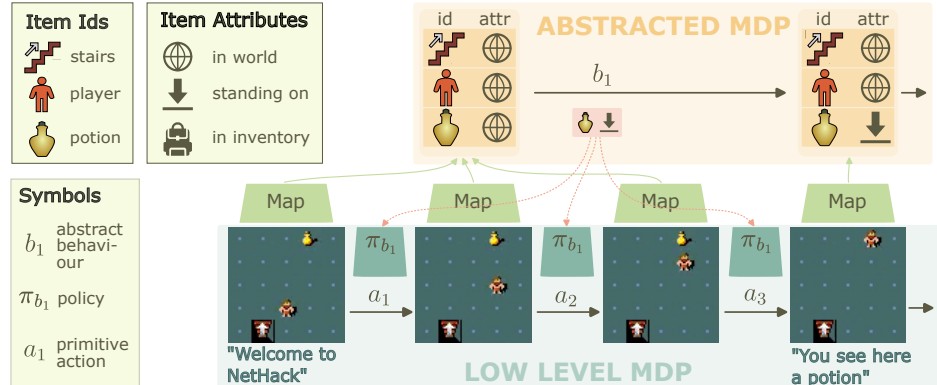

Figure 1: An example state transition in an Ab-MDP (here defined in MiniHack). Abstract states are sets of (item identity, item attribute), and behaviour $b_1$ have structure (item, new attribute). An abstract state can correspond to multiple low level states, and an abstract behaviour multiple primitive actions. We provide legends for the item identities and attributes illustrated (left rectangles).

## 3 METHODS

While any standard RL methods can be used to solve the Ab-MDP, we develop a method to make better use of the item-attribute structure and item-perturbing behaviours. Specifically, we design a model-based method that efficiently explores and learn an accurate full world model. Our method, **M**odel-based **E**xploration of abstracted **A**ttribute **D**ynamics (**MEAD**) can be broken down into three components: (i) A forward model $f_\theta(X, b)$, (ii) A reward function $r(X, b)$, (iii) A planner which maximizes the reward function.

Our forward model takes in the current abstract state $X$ and behaviour $b$, and predicts the *probability of success* of the change proposed by the abstract behaviour. We describe the procedure for training the forward model in Section 3.1.

We use a different reward function and planner at training versus inference time. During training, the agent learns an accurate (abstract) world model without rewards. We use an intrinsic reward function $r_{\text{intr}}(X, b)$ that encourages infrequently visited states, and plan using Monte-Carlo Tree Search (MCTS) toward these states. Section 3.2 describes the exploration phase in detail.

At inference time, our aim is to reach a particular abstract state. We define our reward function $r_{\text{goal}}$ as 1 for the goal abstract states and 0 for all others. Because our forward model has been trained to accurately estimate state transitions, we simply run Dijkstra's algorithm using $f_\theta$ and $r_{\text{goal}}$ to maximize expected reward. Section 3.3 describes this.

## 3.1 FORWARD MODEL

A forward model takes in a current abstract state and behaviour to model the distribution over next states, $X_{t+1} \sim p(X_{t+1}|X_t, b_t)$. Typically, this distribution has support over the full state space $\mathcal{X}$. We observe that by construction, Ab-MDP considers competent policies as ones that change *single* item-attributes. Thus, we assume after an abstract transition, either (i) a single item's attribute changes according to $b_t$, or (ii) the abstract state is unchanged. It is of course possible for multiple items' attributes to change at once, or for an unexpected item's attribute to change—we show with a competent set of policies and re-planning our model is robust to this in practice in Appendix B.2.2.

**Discriminative modelling** Given an abstract transition $(X_t, b_t, X_{t+1})$, we consider this transition **successful** if the next abstract state contains the attribute change proposed by the behaviour: $(\alpha^{(i)}, \xi') \in X_{t+1}$ for $b_t = (\alpha^{(i)}, \xi')$. *Executing a competent policy has a high probability of leading to a successful transition.*

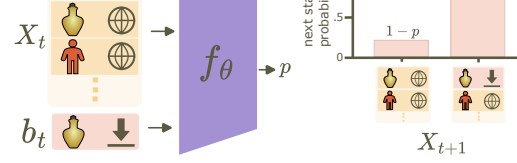

We model this **success probability** directly,

$$q(X_t, b_t) = \Pr(x' \in X_{t+1}|X_t, b_t), \quad (1)$$

$$\text{where } x' = b_t = (\alpha^{(i)}, \xi').$$

Figure 2: The forward model $f_\theta$ predicts the probability $p$ that the behaviour $b_t$ is successful from state $X_t$. The next state distribution is modelled as a binary distribution (Equation 3).

We refer to this way of modelling as **discriminative world modelling**, which models the probability that the state change specified by a behaviour is successful. As behaviours are by construction item perturbations, we can view this as asking a question about allowable item changes ("can I change this item to have this attribute?"). This is different from *generative world modelling* which models $\Pr(X_{t+1}|X_t, b_t)$ without any restrictions on possible future states (i.e. "what are all possible outcomes of my action in this state"). We find this inductive bias leads to more data efficient learning and multi-step planning (Section 4.4.1).

**Forward Prediction** We can model the next state after a *successful* transition as,[2]

$$X' = \Delta(X, b) = X \setminus x^{(i)} \cup (\alpha^{(i)}, \xi'), \quad \text{for any } b = (\alpha^{(i)}, \xi'). \quad (2)$$

Then, given success probability function $q$ (Equation 1), we model the next state distribution as:

$$\Pr(X_{t+1}|X_t, b_t) = \begin{cases} \Delta(X_t, b_t) & \text{with probability } q(X_t, b_t), \\ X_t & \text{with probability } 1 - q(X_t, b_t). \end{cases} \quad (3)$$

See Figure 2 for an example. As mentioned above, our model is robust to cases where multiple items change through re-planning (Appendix B.2.2).

**Model Learning** We use model function class $f_\theta : \mathcal{X} \times \mathcal{B} \rightarrow [0, 1]$ with trainable parameters $\theta$, and fit it to approximate success probability (Equation 1). We can estimate the empirical success probability from a dataset of trajectories simply by counting, and fit a binary cross entropy loss. We found this to be a stable, simple-to-optimize objective. More details in Appendix C.1.

## 3.2 PLANNING FOR EXPLORATION

We want the agent to explore indefinitely to discover how the world works and represent it in its model $f_\theta$. We use a count-based intrinsic reward introduced in Zhang et al. (2018) which encourages the agent to uniformly cover all state-behaviours (details in Appendix C.3):

$$r_{\text{intr}}(X, b) = \sqrt{T/(\text{N}(X, b) + \epsilon T)}, \quad (4)$$

where $\text{N}(X, b)$ is a count of the number of times $(X, b)$ is visited, $T$ is the total number of (abstract) time-steps, and $\epsilon = 0.001$ a smoothing constant.

We use Monte-Carlo Tree Search (MCTS) to find behaviours to reach states that maximizes Equation 4. This is done within the partially learned model's imagination and allow the agent to plan toward novel states multiple steps in the future. The agent takes the first behaviour proposed. Further, since behaviours do not always succeed, one needs multiple state-visitations to estimate $q(X, b)$ well. We found MCTS exploration discovers more novel transition than exploring randomly (Section 4.4.2).

---

[2]We use set notations to denote removal of the item $x^{(i)} = (\alpha^{(i)}, \xi^{(i)})$ from the set $X$, and replacing it with new item $(\alpha^{(i)}, \xi')$.

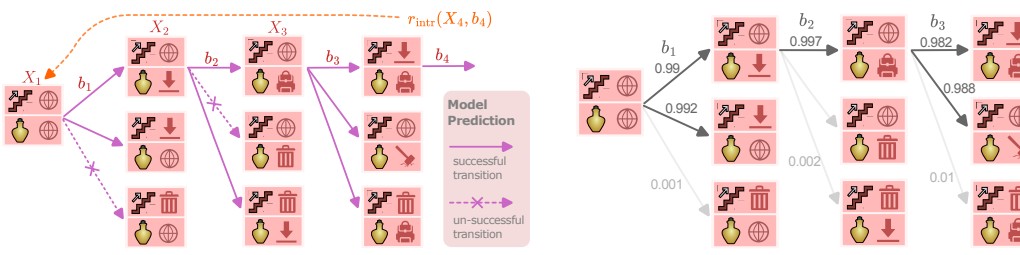

(a) Task agnostic training using MCTS and a count based reward to explore and learn the world model

(b) Dijkstra planning at eval time to maximize success probability of reaching any goal state

Figure 3: Planning within model's imagination to both explore, and to reach any goal state.

### 3.3 PLANNING FOR GOAL

Once the agent has sufficiently explored the environment and is well-fitted, the model $f_\theta$ usually contains a small set of high success probability edges between abstract states (see Figure 3b; most item-attribute changes are generally impossible). We can use this model $f_\theta$ to plan to reach any desired (abstract) world state. As our model is not trained to maximize external rewards, we simply give it a function that tells it when it is in a desired state: $r_{\text{goal}}(X) = 1$ if $X$ is a goal state, and 0 otherwise.

We use Dijkstra's algorithm which finds the shortest path in a weighted graph. We turn our world model into a graph weighted by success probabilities, using a negative log transform:

$$d(X, \Delta(X, b)) = -\log f_\theta(X, b), \quad \text{with } \Delta(X, b) \text{ defined in Equation 2.} \quad (5)$$

Thus, finding the shortest path between current and goal state returns an abstract behaviour sequences that maximizes the probability of successfully reaching the goal.

### 4 RESULTS

We use two sets of of environment. One is 2D crafting, which requires agent to craft objects following a Minecraft-like crafting tree with each crafting stage requiring multiple low level actions (Chen et al., 2020). Second is MiniHack (Samvelyan et al., 2021), which contains a series of extremely difficult exploration games (Henaff et al., 2022). We detail the environments further in Appendix D.

For baselines, we use strong model-free (asynch PPO, (Schulman et al., 2017; Petrenko et al., 2020)) and model-based (Dreamer-v3 (Hafner et al., 2023; Sekar et al., 2020) and MuZero (Schrittwieser et al., 2020; Werner Duvaud, 2019)) baselines. Dreamer-v3 learns a generative world model and plans using model-free method in imagination, while MuZero learns a generative world model and plans using MCTS. We outline different baseline methods in greater details and do additional comparisons in Appendix E. All the methods are trained in the Ab-MDP. We report low level steps and plot the 95% confidence interval. See Appendix G for other experimental details.

### 4.1 LEARNING FROM SCRATCH IN SINGLE ENVIRONMENTS

Figure 4 shows the from-scratch Ab-MDP training results for strong baselines and our method. We see that Dreamer-v3, a state-of-the-art model-based method, is highly sample efficient. MuZero, another model-based method utilizing online planning, is less efficient. PPO, a model-free method, stably optimizes the reward, but at a data budget of an additional $\sim$ two orders of magnitude (hundreds of thousands versus 10 million). Our method learns the Ab-MDP with an additional order of magnitude sample efficiency improvement compared to the already efficient Dreamer. We also refer the reader to Figure 3b (which is a clean illustration of the same data presented in Appendix F.6) for an example of the interpretable world model we can extract from MEAD.

For reference, we plot the performance for SOTA algorithms in the low level MDP (without Ab-MDP) as triangles and stars. In particular in MiniHack tasks (Fig.4b), model-free methods with random and global exploration bonuses (IMPALA, RND, ICM) perform poorly even after 50 million steps, and methods using episodic exploration bonus perform better (E3B). Nonetheless, all of them are less efficient than the least efficient method trained on the Ab-MDP.

These results show that Ab-MDP helps simplify the problem, with all methods reaching reasonable performances—a feat not achievable by most methods trained in the low level MiniHack MDP

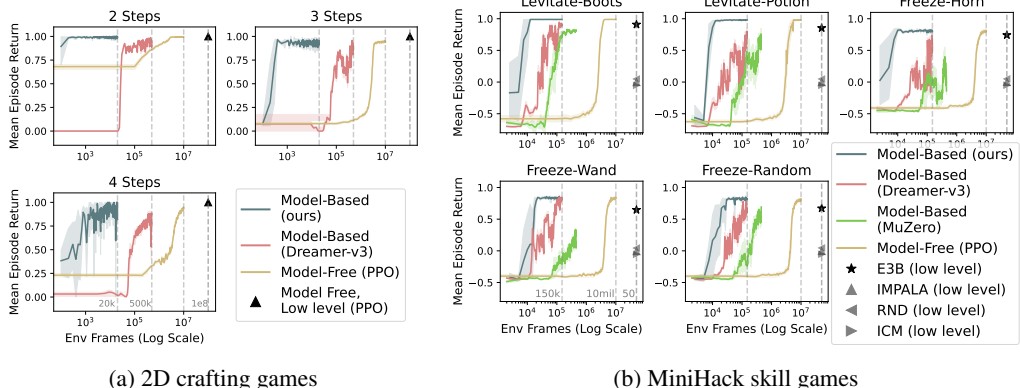

(a) 2D crafting games          (b) MiniHack skill games

Figure 4: Results in games trained from scratch in Ab-MDP. Triangles and stars denote low-level only methods. For MiniHack, we also show the final performance of state-of-the-art exploration methods in the low level MDP (star/triangle), from Henaff et al. (2022). **X-axis on log scale**.

(Henaff et al., 2022). However, we also see that the problem is still non-trivial, with the model-free baseline still requiring millions of frames to achieve success. Our method reaches the same peak performance as the model-free method while using fewer samples than the model-based baseline.

## 4.2 TRANSFER AND COMPOSITIONS

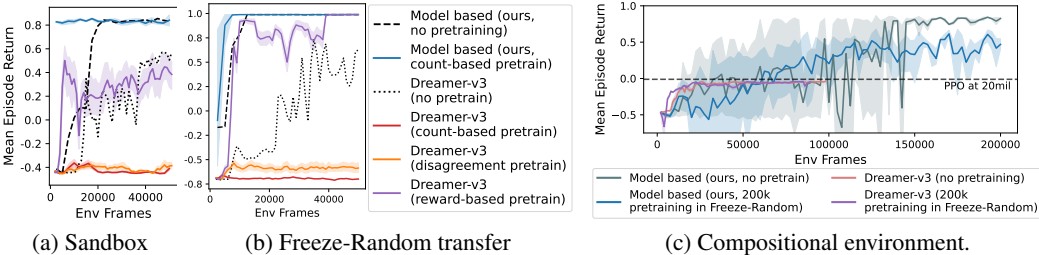

(a) Sandbox      (b) Freeze-Random transfer      (c) Compositional environment.

Figure 5: Transfer experiments and a difficult compositional environment. Dotted lines in 5a and 5b are the same curves from Figure 4b, with just the means shown for clarity.

Next, since the Ab-MDP abstraction models object identities and shared attributes, we evaluate if the abstract level models can transfer object knowledge to new settings. Here we exclusively use the MiniHack environments due to their rich repertoire of object types and higher difficulty. Note we run each transfer experiments in multiple environments; as the result is similar, we report a single illustrative environment in Figure 5 for brevity and provide full results in the Appendix Figure 16.

**Sandbox transfer** We first place the agent in a "sandbox" environment where *all* objects are spawned with some probability (Appendix D.2.1). The agent is allowed to interact with the environment with only an exploration reward (no task-specific reward) for 200k frames. We then fine-tune the agent in each of the environments. For our model, we continue to train $f_\theta$ within the new environment. For Dreamer-v3, we transfer only the world model (encoder, decoder and dynamics function) and re-initialize the policy and reward functions.[3] Results for all five environments follow similar trends; we show a representative example in Figure 5a, with full results in Figure 16a.

Surprisingly, Dreamer-v3 trained with both count- and disagreement-based intrinsic rewards fails to get any meaningful rewards in the allocated fine-tuning budget (Figure 5a, red & orange curves), although training for longer seems to show a some reward—indicating negative transfer (see Appendix F.1.2). Similarly, the model-free PPO policy shows negative transfer—reaching lower performance with the same data budget as compared to training from scratch (results in Appendix F.1.1 since x-axis in different order of magnitude). A possible explanation is while the agent observes all object types, the data distribution is dominated by seeing *more* objects on average in the training

---

[3]The point of this experiment is to evaluate world knowledge transfer. Nevertheless, fine-tuning the exploratory reward function and policy also achieves no better performance.

environment than in the evaluation environments, thus presenting a distribution shift. In contrast, our model exhibits good zero-shot performance, and reaches optimal performance efficiently.

To help the Dreamer-v3 baseline further, we give it a reward function to "use" the items seen in the pre-training sandbox environment. This is a kind of privileged information which biases the model representation to better model the evaluation-time task. We observe that this variant of Dreamer-v3 exhibits positive transfer, hinting at Dreamer's representation reliance on the reward function. Nevertheless, it does not exhibit the same degree of zero-shot or fine-tuning capability as our model.

**Transfer across object classes** We further evaluate the ability of the models to transfer to new object types. For example, would a model learn to use a potion better, after it has learned to use a freeze wand, since the abstract level interactions of these objects are similar (the agent can stand on top of these objects, pick it up, and use them)? We take 200k frame checkpoints in the `Freeze Random` environment and train them in the other 4 environments for an additional 50k frames.

We again show one representative comparison in Figure 5b and present the full 4-environments result in Figure 16b. We see our model exhibits reasonable zero-shot performance and improves further through fine-tuning. Similar to the sandbox environment, we observe that exploration-based training of both PPO (results plotted in Appendix F.1.1 due to x-axis being in different order of magnitude) and Dreamer-v3 exhibits negative transfer in our setting, with both methods eventually achieving better rewards but slower than training from scratch.[4] We again pre-train Dreamer-v3 with privileged reward information that matches the downstream task, but still requires generalization to novel objects. We see that in general this way of training dreamer resulted in positive transfer – it learns to reach a higher reward with fewer samples compared to training from scratch.

It is somewhat surprising that our model transfer so well zero-shot to completely new objects in Levitate-Boots and Levitate-Potion. We analyze this further in Appendix F.5: for (item id, item attribute) vectors with unseen ids, we find the model internally cluster them by object attributes, which generalizes from Freeze objects to Levitate objects.

**Compositional planning environment** Finally, we designed an environment similar to the 5 difficult Minihack exploration levels, but with the required planning depth twice as long.[5] This is an even more challenging task where the agent needs to pick up both a freeze and a levitate object, make itself levitate (using the levitate object), then cast a freezing spell to terminate the episode. Further, the levitate spell must be cast *after* all the objects have been picked up (since the agent cannot pickup objects while levitating), but before the freeze spell is cast. This is a challenging environment requiring precise sequences of abstract behaviours.

We observe in Figure 5c that when pre-trained on Freeze objects, our model trains more stably in this environment (but no longer shows good zero-shot results), succeeding majority of times. Our from scratch model solves this task with better rewards. Our method also learns a highly complex yet still interpretable world graph, with a simplified version shown in Figure 34. In contrast, all other methods, with and without pre-training, do *not* find the correct solution at all, but instead are stuck at a 0 reward local minima.

### 4.3 LEARNING OBJECT PERTURBING POLICIES AND OBJECT-CENTRIC MAP

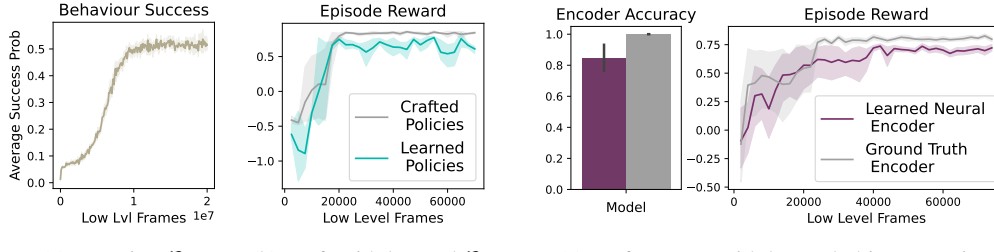

(a) Learning $\mathcal{B}$      (b) Perf. with learned $\mathcal{B}$      (c) Performance with learned object mapping

Figure 6: Performance using learned Ab-MDP components. All error bars denote 95 confidence interval of mean with exception of encoder accuracy which shows 1 standard deviation.

---

[4]PPO and extended Dreamer-v3 curves in Appendix F.1.

[5]Previous environments need a minimum of three abstract transitions to complete; this one at least six.

Up to this point we have assumed the *existence* of an object-centric mapping and competent behaviours (Sec. 2.1), which can be combined with our method for efficient exploration and world model learning. We now demonstrate how one might *learn* both components.

### 4.3.1 LEARNING LOW LEVEL POLICIES

We observe low level policies can be learned with RL if an object-centric map exists. Specifically, we reward the agent for achieving an abstract transition *successfully*: from $X_t$, if executing behaviour $b = (\alpha^{(i)}, \xi')$ until an abstract transition results in $X_{t+1}$ containing item $(\alpha^{(i)}, \xi')$, the associated policy $\pi_b$ gets a reward of +1 (else -1). We train low level policies to replace all behaviours in the MiniHack single task environments, and observe the success probability can be stably optimized with PPO (Fig.6a; note it does not reach 1 as it is averaged over all attempted behaviours, some are for impossible transitions). Solving Ab-MDP with learned behaviours perform similarly (Fig.6b). The reported 2D Crafting results (Section 4.1) already use learned behaviours. Details in Appendix F.2.

### 4.3.2 LEARNING OBJECT-CENTRIC MAP

If we wish to also learn the object-centric mapping, we can do so using supervised learning. We generate a dataset of 100k transitions using a random policy in MiniHack's `Freeze-Horn` environment, and use the hand-crafted mapping to provide ground truth abstract state labels. We train a neural net to produce the abstract state $X$ from $S$. We compare planning performance using the learned object-centric mapping in Figure 6c and observe it reaches similar performance as the ground truth mapping, even though the encoder is not perfect in predicting the abstract states. We discuss alternative way of getting this mapping in Section 6.

### 4.4 ABLATIONS

We ablate major design choices below, with additional ablations in Appendix F.3.

### 4.4.1 GENERATIVE VS. DISCRIMINATIVE MODELS

A common way of learning a forward model is to model the next state *generative-ly*, i.e. $\hat{X}_{t+1} \sim f(X_t, b_t)$, with support over the entire state space $\mathcal{X}$. In MEAD, we opted to constrain the possible future to either be the same as the current time-step, or one where there is a single item change (Fig. 2). This has the disadvantage of being less flexible, but the advantage of being easier to model. To isolate only the effect of discriminative modelling, we design an experiment where we fit models to the same expert dataset, with the

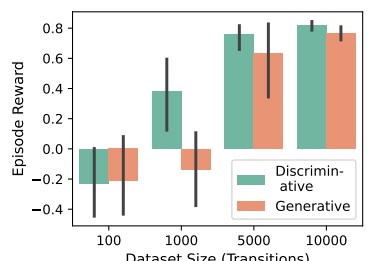

Figure 7: Planning performance with discriminative vs. generative models

*only* difference being the model type (fit to either predict success probability in the discriminative case, or to predict the distribution over next item-attribute sets), while keeping all other variables constant. Indeed, given the same (expert) data budget, a discriminative model learns a better model for Ab-MDP planning in the lower data regime (Figure 7, experimental details in Appendix F.3.1).

### 4.4.2 COUNT-BASED EXPLORATION

We observe using the count-based intrinsic reward (Equation 4) with MCTS was important for exploration. Figure 8 shows count-based exploration discovering many more valid object attribute transitions compared to a randomly exploring agent. Correspondingly, this leads to a model $f_\theta$ that is better at achieving goals.

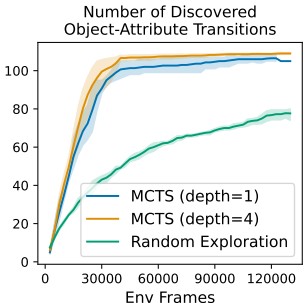

Figure 8: Effect of count-based exploration in the Freeze-Wand environment. MCTS discovers more unique valid (item identity, item attribute, new attribute) transitions.

### 4.4.3 PARAMETRIC VS. NON-PARAMETRIC MODELS

Instead of fitting a discriminative model to success probability, we can directly use the empirical count-based estimate from a dataset (a non parametric model). This is done to learn the model in Zhang et al. (2018). We compare this choice in Appendix F.1.3 and show our parametric model has better sample efficiency, demonstrating the generalization benefits of our parametric forward model.

## 5 RELATED WORK

**Hierarchical RL**    Our Ab-MDP is a hierarchical state and temporal abstraction formalism and can be interpreted under the Options Framework (Appendix B.1). Within the Options Framework (Sutton et al., 1999; Precup, 2000), Option discovery remains an open and important question. Existing methods typically do option discovery in a "bottom-up" fashion by learning structures from their training environments (Bacon et al., 2016; Machado et al., 2017; Ramesh et al., 2019; Machado et al., 2021). In contrast, our Ab-MDP uses "top-down" prior knowledge in the form of object-perception, and define options that perturb objects' attributes.

**Planning with abstract representations**    Abstraction is a fundamental concept (Abel, 2022) and we select a subset of most relevant works. The Ab-MDP can be viewed as a more structured MDP formulation in which learning and planning is simplified. This relates to the Factored MDP (Boutilier et al., 1995; 1999; Guestrin et al., 2003; Degris & Sigaud, 2013), which models the conditional independence structure between factors in a factorized state space as Bayesian Networks. It also relates to OO-MDP (Diuk et al., 2008; Keramati et al., 2018), which describes state as a set of object and object states which interact through relations. We discuss their mathematical relationships in Appendix B.1.3. Also related is MaxQ and AMDP (Dietterich, 1998; Gopalan et al., 2017), which provides a decomposition of a base MDP into multi-level sub-MDP hierarchies to plan more efficiently. Transition dynamics can also be defined as symbolic domain specific language (Mao et al., 2023). Above works require pre-specifying additional interaction rules, functions, and object relations to simplify planning, whereas we only assume the ability to see objects and their attributes, and learn the forward model as a neural network. The CEE-US model of Sancaktar et al. (2022) use ensembles of graph neural networks to exploit an object-centric state representation which factorizes into entity and agent information, and show strong model learning and zero-shot generalization capabilities in a robot manipulation domain. DAC-MDP (Shrestha et al., 2020) clusters low level states into "core states" to solve by value iteration, with a focus on getting good policies in an offline RL setting rather than abstract world model learning. More distantly related is Veerapaneni et al. (2019) which focus more on learning good object-centric representations rather than effective usage of an abstract level. Nasiriany et al. (2019) tackles sub-policy learning and planning with abstraction, focusing on implicitly representing abstractions rather than using semantically meaningful ones.

Zhou et al. (2022); Haramati et al. (2024); Chang et al. (2023) are works that leverage object-centric representations with a set of discrete, factorized descriptions—a beneficial inductive bias Ab-MDP shares (i.e. item identities and attributes). In particular, Chang et al. (2023) uses slot attention to learn factorized "entity types" and "entity states", albeit relying on more restrictive assumptions about entity dynamics. Haramati et al. (2024) extracts an object-centric representation using *Deep Latent Particles*, which extracts latent "particles" describing pixel-based objects in terms of their pixel location, size, etc. Both works offer potential ways of learning the object-centric map M. We discuss their relationship with Ab-MDP further in Appendix B.1.3 and compare against the model-learning method from Chang et al. (2023) in Appendix F.1.3.

Most similar is Zhang et al. (2018), which does model-based planning over abstract attributes. We differ in having a more expressive abstraction framework: modelling a flexible set of objects and shareable attributes, rather than a fixed, binary set of attributes only. We also use a parametric model to more efficiently learn and is able to transfer object information to new environments. We compare against their non-parametric approach and show better sample efficiency in Section 4.4.3.

**RL in semantic spaces**    As language models have grown in interest and sophistication, there has been an increasing interest in semantic and linguistic abstractions in RL. Early works such as Andreas et al. (2017) looked at handcrafted abstract spaces such as we do here to show that creating abstractions allows for better learning and exploration. Other works such as Jiang et al. (2019); Chen et al. (2021) specifically use natural language as the abstract space. In others, the language space abstraction is used as a way of guiding exploration (Tam et al., 2022; Guo et al., 2023).

**Model-based RL**    Our work is related to other work in Model-based Reinforcement Learning (Sutton, 1990; Kaiser et al., 2019; Doya et al., 2002). In particular we compare to the Dreamer line of work (Hafner et al., 2019; 2023; Sekar et al., 2020; Mendonca et al., 2021; Hafner et al., 2022). These methods were developed for a pixels setting and employ a two step learning strategy: learning a generative forward model to predicts the next states $X'$, then learning a policy in imagination. We compare directly to this in our experiments, where we show that our method (which employs a discriminative forward model) learns more quickly and transfers better in our setting.

**Exploration in RL**  Exploration is a well-studied problem in RL. For a more extensive literature review, Amin et al. (2021) gives a comprehensive picture. Most relevant to here are count-based methods such as Raileanu & Rocktäschel (2020); Zhang et al. (2021). Our work takes inspiration from the hard exploration problems from Henaff et al. (2022). There are also many disagreement-based methods including Sekar et al. (2020) and Mendonca et al. (2021); we compare directly to the former for our Dreamer benchmarks with pre-training.

## 6 DISCUSSION

In this work we introduce the Ab-MDP, which uses human understandable object-attribute relations to build an abstract MDP. This enables us to learn a discriminitive model-based planning method (MEAD) which explores and learns this semantic world model without extrinsic rewards.

**Limitations** The major limitation of the work is that the construction of Ab-MDP requires the existence of an object-centric mapping. While we show how these components can be learned in Section 4.3, learning the object mapping still requires a labelled dataset, and a method to automatically discover object-centric representations in the form of items and attributes remains an open question. Promisingly, the field as a whole is moving in a direction where abstractions are becoming more easily obtainable. For instance, object-centric abstract maps can be acquired using segmentation models (Kirillov et al., 2023) and visual language models (Alayrac et al., 2022). Similarly, the short-horizon (low-level) manipulation policies can be acquired more efficiently through imitation learning. For example, it has been found in robotics that simply using off-the-shelf models for perception and skill policies is possible (Liu et al., 2024). Finally, we currently focus only on *discrete* attributes, meaning the Ab-MDP cannot precisely capture continuously varying quantities (e.g. position and velocity). However, this is only a restriction in the *abstract level*, and does not impose any fundamental restriction on the kind of learnable policies in the *base* MDP. In other words, Ab-MDP models discrete, abstract relationships present in the base MDP, while low-level policies remain free to utilize continuous features and output continuous actions in the base MDP to drive item-attribute changes for the Ab-MDP.

Overall, as foundational models advance, the field is progressively moving toward working with discrete, semantically meaningful representations. Concurrently, reinforcement learning provides rigorous and principled frameworks for designing intelligent agents, encompassing exploration, planning, and hierarchical behaviours. We hope this work demonstrates a productive synthesis of these perspectives, and takes a step forward in the development of more intelligent autonomous systems.

ACKNOWLEDGMENTS

This work is supported by ONR MURI #N00014-22-1-2773, ONR #N00014-21-1-2758, and the National Science Foundation under NSF Award 1922658. AGC is supported by the Natural Sciences and Engineering Research Council of Canada (NSERC), PGSD3-559278-2021. *Cette recherche a été financée par le Conseil de recherches en sciences naturelles et en génie du Canada (CRSNG), PGSD3-559278-2021.* We are grateful for insightful discussions with Rajesh Ranganath. We thank Doina Precup, Arun Ahuja, Ben Evans, Siddhant Haldar, Nikhil Bhattasali, Ulyana Piterbarg, Nikhil Dongyan Lin, and Mandana Samiei for their helpful feedback on earlier drafts of this paper. Finally, we thank the anonymous reviewers for making this work stronger in many ways.

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

## A  BROADER IMPACT

Our contributions are fundamental reinforcement learning algorithms, and we hope our work will contribute to the goal of developing generally intelligent systems. However, we do not focus on applications in this work, and substantial additional work will be required to apply our methods to real-world settings.

Regarding compute resources, we use an internal clusters with Nvidia A100 and H100 GPUs. All experiments use at most one GPU and are run for no more than two days. Running baselines along with randomization of seeds requires multiple GPUs at once.

## B  PROBLEM SETTING DETAILS

### B.1  ABSTRACTED MDP

Here we re-iterate our framework and discuss it in greater details. We first define a low level (reward-free) Markov Decision Process (MDP, Puterman (1994)) as the tuple $\langle \mathcal{S}, \mathcal{A}, P \rangle$, with (low level) state space $\mathcal{S}$, primitive action space $\mathcal{A}$, and transition probability function $P : \mathcal{S} \times \mathcal{A} \times \mathcal{S} \to [0, 1]$. Suppose the agent interacts with the environments at discrete time-steps $u = 0, 1, 2, ...$, then $P(s_{u+1}|s_u, a_u)$ describes the probability of transitioning to state $s_{u+1} \in \mathcal{S}$ after one time-step when choosing action $a_u \in \mathcal{A}$ in the current low level state $s_u$.

To construct an Ab-MDP, we start with a surjective deterministic mapping M from low to abstract level: $M : \mathcal{S} \to \mathcal{X}$. That is, any low level state $S \in \mathcal{S}$ maps onto some abstract state $X \in \mathcal{X}$ (multiple $S$ can and do map to the same $X$).

Define behaviours $\mathcal{B}$. Each $b = (\alpha^{(i)}, \xi') \in \mathcal{B}$ describes a single item id $\alpha^{(i)}$ and a desired *new* attribute, along with an associated low level policy $\pi_b : \mathcal{S} \to \mathcal{A}$ which tries to set the specified item to have the new attribute. A behaviour can be competent or incompetent. Executing competent behaviours lead to the specified item-attribute changes with high probability.

A behaviour may be incompetent because (i) the proposed attribute change is impossible within the rules of the world (e.g. item is not a craft-able object) (ii) the attribute change is not possible from the current state (e.g. do not have the correct raw ingredients in inventory), or (iii) the behaviour is not sufficiently trained to perform the specified change.

We build a simple abstraction that allow us to learn purely at the abstract level while still being able to influence actions at the low level. Notice the surjective map M give us *state abstraction*. We build *temporal abstraction* by defining abstract state transitions as happening at a slower time-scale than the low level time-step $u$.

**Definition B.1.** An **abstract state transition** has occurred at (low level) time-step $u$ if: (i) the abstract state has changed, $\mathcal{M}(s_u) \neq \mathcal{M}(s_{u-1})$; or (ii) the abstract state has not changed in the past $k$ steps, $\mathcal{M}(s_u) = \mathcal{M}(s_{u-1}) = ... = \mathcal{M}(s_{u-k})$.

**Definition B.2.** An **(max-k) Ab-MDP** is defined as the tuple $\langle \mathcal{X}, \mathcal{B}, \mathbf{T}_k \rangle$, with abstract states $\mathcal{X}$, abstract behaviours $\mathcal{B}$, and transition probability function $\mathbf{T}_k : \mathcal{X} \times \mathcal{B} \times \mathcal{X} \to [0, 1]$. $\mathbf{T}_k(X'|X, b)$ describes the probability of being in abstract state $X'$ when starting from $X$ and executing behaviour $b$ *until an abstract state transition occurs* (Definition B.1).

In the main text, we write *Ab-MDP* and denote the transition function as $\mathbf{T}$ for brevity (instead of writing max-k Ab-MDP and transition function $\mathbf{T}_k$). $k = 8$ was chosen for baseline methods as it worked slightly better. MEAD works just as well with $k = 8$ and $k = 16$.

#### B.1.1  INTERPRETATION VIA OPTIONS FRAMEWORK

It is natural to interpret our hierarchical framework within the Options framework (Sutton et al., 1999). An option consists of a policy $\pi : \Omega \times \mathcal{A} \to [0, 1]$, a termination function $\beta : \Omega \to [0, 1]$, and an initiation set of states $\mathcal{I} \subseteq \mathcal{S}$. $\Omega$ is used to denote the set of all possible historical states.

Our set of behaviour $b \in \mathcal{B}$ *is* a set of options. Each behaviour specifies a behaviour policy $\pi_b$, which deterministically terminate when an abstract state transition occurs (Definition B.1), and they can be initiated from all states ($\mathcal{I} = \mathcal{S}$).

The point of the Options Framework and temporal abstraction generally is to break up a monolithic problem into re-usable sub-problems. *How* one breaks up a problem is an open question. Some popular previous approaches include finding bottleneck states and using the successor representation to discover well-connected states. We can view our approach as another way of breaking up the problem *when prior knowledge in the form of objects is available*: we break up the problem at the level of single item-attribute changes (with options defined as policies that can change a single item's attribute).

### B.1.2 A NOTE ON SEMI-MDP

The decision process that selects amongst options is a semi-MDP (Theorem 1, (Sutton et al., 1999)). The general way to model semi-MDP dynamic requires considering the amount of low level timesteps that passes between each abstract level decision as a random variable (Sutton et al., 1999; Barto & Mahadevan, 2003). Instead, we chose to simply treat the abstracted level as having a fixed time interval in between steps. This is chosen for simplicity as our focus is with learning efficiency on the abstract level, and this formulation allow us to directly apply efficient algorithms developed to solve MDPs. We also see empirically that given our $\mathcal{M}$ and $\mathcal{B}$ this choice does not lead to a less optimal policy compared to algorithms that directly solve the low level MDP.

### B.1.3 RELATIONSHIP TO OTHER MDP FORMALISM

Our Ab-MDP is related to **Factored MDPs** (Boutilier et al., 1995; 1999; Degris & Sigaud, 2013). Factored MDPs model states as decomposable into a set of factors $X_t = \{x^{(1)}, x^{(2)}, ...\}$, with each factor having domain $\text{Dom}(x^{(i)})$. This allows the transition probability function to be decomposed into Dynamic Bayesian Networks (DBNs) between the factors at two successive time-steps $t-1$ and $t$, with the factors as the nodes of the DBN. Concretely, this decomposes the transition probability function for each action into a product of conditional probabilities,

$$P_a(X_t|X_{t-1}) = \prod_i P_a(x_t^{(i)}|\text{Parent}(x_t^{(i)})), \qquad (6)$$

where $\text{Parent}(x^{(i)}))$ denotes the factors in $X_{t-1}$ which are the parent of $x_t^{(i)}$. In the most generic form, each action $a \in \mathcal{A}$ induces a different DBN. If the DBN is known or well-approximated, the Factored MDP can be efficiently solved (Guestrin et al., 2003).

Similarly, the Ab-MDP considers a decomposition of state into items (analogous to factors), and our method MEAD shares the philosophy of efficient utilization of a factorized state space. While one could interpret Ab-MDP as a kind of Factored MDP with discrete domains, the two differs in their emphasis on transition vs. actions. Factored MDPs defines arbitrary actions, each inducing a potentially different conditional independence structure in the transition dynamics. The Ab-MDP instead defines a set of *next state changes* as the actions, without explicitly specifying any independence structures in its transitions dynamics. Our method MEAD then makes use of this action representation to directly model the probability of item-attribute changes, while implicitly learning the transition dynamics in the weights of the neural network.

Our Ab-MDP is also related to the **Object-oriented MDP (OO-MDP)** of Diuk et al. (2008), and shares a similar structure of describing a set of "objects" and their attributes separately. In OO-MDP, an environment consists of a set of *objects* $O = \{o_1, ..., o_o\}$, each belonging to one of $c$ *classes* $\{C_1, ..., C_c\}$, and each having a set of (class) *attributes* $\text{Att}(C) = \{C.a_1, ..., C.a_a\}$. An OO-MDP "object state"—$o$.state—is a value assignment to all of its attributes ($\{a_1, ..., a_a\}$), and the OO-MDP "state" is the union of all of its object states: $s = \cup_{i=1}^o o_i.\text{state}$.

Our Ab-MDP similarly contains a set of *items*, $X = \{x^{(1)}, ..., x^{(N)}\}$, with each item consisting of an *item identity* and an *attribute*, $x^{(i)} = (\alpha^{(i)}, \xi^{(i)})$. We can view our Ab-MDP as a soft generalization of OO-MDP by modelling:

- The OO-MDP *object* (and its associated class) as an Ab-MDP *item identity*: $\alpha^{(i)} = o_i$,
- The OO-MDP *attribute* as the Ab-MDP *item attribute*: $\xi^{(i)} = \{C.a_1, ..., C.a_a\}$, $o_i \in C$.

To model state *transitions*, OO-MDP makes additional assumptions that objects interact through their class identities. Specifically, a set of *relations* determine if two objects interact, with resulting

*effects* that is used to update the object state. This requires the designer to additionally provide a set of accurate relations functions and effects, with the benefit of facilitating within-class generalization. In contrast, Ab-MDP makes no additional assumption about object class, relations, or interactions between objects. We instead directly models (probabilistically) whether an *effect* is possible. This is done in a fully data-driven way through a neural function $f : \mathcal{X} \times \mathcal{B} \rightarrow [0, 1]$ and empirically is shown to generalize well across item-attribute variations. Item interactions are implicitly represented by the attention mechanism over the set of input items.

The **neural constraint satisfaction** (NCS) approach of Chang et al. (2023) introduces an *entity-set* state representation, where each *entity* $h^{(i)}$ is decomposable into *type* $z^{(i)}$ and *entity state* $s^{(i)}$. Our Ab-MDP can naturally be mapped onto their entity-state abstraction by setting an Ab-MDP *item identity* as the NCS *type* ($\alpha^{(i)} = z^{(i)}$) and the Ab-MDP *item attribute* as the NCS entity state ($\xi^{(i)} = s^{(i)}$). Note the Ab-MDP "behaviour" (i.e. abstract action) is equivalent to the NCS "action".

NCS makes additional assumptions that attribute change is dependent only the behaviour: $\xi_{t+1}^{(i)} = f(\xi_t^{(i)}, b_t)$. On the other hand, Ab-MDP models more general relationships $\xi_{t+1}^{(i)} = f(\alpha_t^{(1)}, \xi_t^{(1)}, ..., \alpha_t^{(N)}, \xi_t^{(N)}, b_t)$ (e.g. Equation 1). The NCS assumption is restrictive in that it would fail to capture cases where an item's attribute change depends on what the item's identity is, and/or the attributes of other items.

The **Deep Latent Particles** (DLP) approach used in Haramati et al. (2024) learns a latent space containing a set of $K$ *particles*: $z = \{(z_p, z_s, z_d, z_t, z_f)_i\}_{i=1}^{K-1}$. $z_p \in \mathbb{R}^2$ describes the particle's (x,y) position on an image, $z_s \in \mathbb{R}^2$ describes its (x,y) scale, $z_d \in \mathbb{R}$ describe its "depth", $z_t \in \mathbb{R}$ its transparency, and $z_f \in \mathbb{R}^l$ the visual features around the particle. We can cast this into the Ab-MDP by having $K$ items whose item identities is the particle's index, $\alpha^{(i)} = i$, and item attribute is a $6 + l$-dimensional vector describing the particle features; $\xi^{(i)} = (z_p, z_s, z_d, z_t, z_f)_i$. DLP can be one way of getting abstraction from pixels, albeit it is restricted to settings where the state is image-based.

### B.1.4 EXAMPLE APPLICATIONS

We provide further examples of possible Ab-MDP constructions for a few common settings. Note we provide one example abstraction per domain, though multiple types of Ab-MDP abstractions can in theory be constructed depending on the choice of the item set and attribute set.

| Environment | Description | Example abstract states | How to get abstract behaviours |
|---|---|---|---|
| Kitchen (Gupta et al., 2019) | A robot arm is in a MuJoCo kitchen with multiple things to do such as turning on lights, turning on kettle, opening oven / microwave / cabinet, etc. | Item set: {light, oven, microwave, cabinet}. Attribute set: {open, closed} | Imitating expert play data in Gupta et al. (2019) which already produce trajectories that lead to abstract state changes |
| Cube pushing (Haramati et al., 2024) | A robot arm pushes cubes on a table to their desired positions | Item set: {red cube, blue cube, green cube, … }. Attribute set: {at goal position, not at goal position} | RL to push a single block to goal at a time |
| Real world robot (Lee et al., 2024) | A Stretch robot in a kitchen, moving objects in and out of containers | Item set: {drawer, bag, oven, box, bread, can}. Attribute set: {open, closed, on table, in bag, in drawer, in oven} | Human imitation learning data containing trajectories between abstract state changes |

Table 1: Example Ab-MDP's applied to various domains

### B.1.5   MULTI-ITEM BEHAVIOURS

In the Ab-MDP, we consider abstract states as an (ordered) set of items $X = \{x^{(1)}, x^{(2)}, ...\}$, where each item consist of an identity and attribute $x^{(i)} = (\alpha^{(i)}, \xi^{(i)})$. A *behaviour* is a description of which item(s) to change (Section 2.1). For instance, the single-item behaviour $b = \{(\alpha^{(i)}, \xi')\}$ specifies a change of the $i$-th item to a new attribute $\xi'$. We can similarly specify multi-item behaviours, such as a two-item behaviour $b = \{(\alpha^{(i)}, \xi'), (\alpha^{(j)}, \xi'')\}$. This behaviour is *competent* if after an abstract state change, the $i$-th item takes on attribute $\xi'$ *and* the $j$-th item takes on attribute $\xi''$ with high probability.

For $N$ objects, $m$ attributes, and $I$ items changes to consider in behaviours,

$$\text{Number of behaviours} = \binom{N}{I} \times m^I . \tag{7}$$

For instance, for single item change behaviours ($I = 1$), there are $\binom{N}{1} \times m^1 = N \times m$ behaviours. For two-item change behaviours ($I = 2$), there are $\binom{N}{2} \times m^2$ behaviours, and so on. In the extreme, one can choose to model a all possible item attribute changes ($I = N$), which results in $m^N$ behaviours that is exponential in the number of items considered.

## B.2   ANALYSIS OF ASSUMPTIONS

### B.2.1   THE GENERALITY OF AB-MDP

We make the observation that *any* sparse reward RL task can be described by a Ab-MDP with a single item,

*Remark* B.3. For any MDP with a reward function $R : \mathcal{S} \to \mathbb{R}$, where,

$$R(S) = 1 \text{ if } S \in \mathcal{G} \text{ and } 0 \text{ otherwise},$$

for $S \in \mathcal{G}$ terminal goal state(s), we can describe this as an Ab-MDP with a single item $\alpha$ and binary attributes $\{\xi_1, \xi_2\}$. The abstract map $\text{M} : \mathcal{S} \to \mathcal{X}$ can be constructed in the following way,

$$\text{M}(S) = \begin{cases} \{(\alpha, \xi_2)\} & \text{if } S \in \mathcal{G}, \\ \{(\alpha, \xi_1)\} & \text{otherwise}. \end{cases} \tag{8}$$

The proxy problem of transition from $X = \{(\alpha, \xi_1)\}$ to $X = \{(\alpha, \xi_2)\}$ solves the sparse reward task in the base MDP and a competent behaviour for this transition is a good policy is the base task.

The point of this is merely to say that Ab-MDP does not restrict the RL problem in a particular way. Of course, such a construction of an Ab-MDP is not interesting and learning a competent behaviour for this Ab-MDP is no easier than learning a behaviour that maximizes the reward function in the base MDP.

The point of Ab-MDP, however, is that a task can be broken-up in a more interesting way, e.g. in our case through a richer set of items and attributes. We demonstrate in this work that an object-centric item-attribute set encoding result in a set of intuitively base behaviours and a world model that is interesting.

### B.2.2   DISCRIMINATIVE MODELLING WITH SINGLE ITEM CHANGE

In our forward model, we made the assumption to only model single item's attribute change (Section 3.1 and Equation 3), and that if the attribute does *not* change then the abstract state $X_{t+1}$ stays the same as $X_t$. We analyze this assumption more deeply in this section.

**Robustness through re-planning**   We first note that in practice, it is often the case that multiple item's attribute changes. For instance, in the 2D Crafting environments, crafting an item often consumes the raw ingredient used to craft it. Thus, a plan that considers multiple items being crafted (attribute change from `absent` to `in inventory`) will inaccurately predict that the raw ingredient stays in the inventory (attribute of raw ingredient stays the same, staying as `in inventory`) rather than changing to `absent`. We illustrate the plan generate at each abstract step in Figure 9, and show that despite this being the case, the plan still generates an optimal action sequence; and that at each time-step, re-planning with the newly obserbved abstract state updates the incorrect abstract prediction to be more accurate.

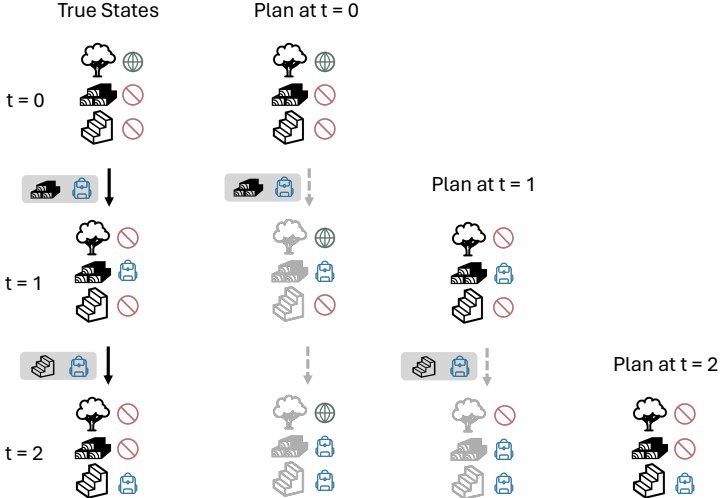

Figure 9: An illustrative example following the logic of the 2D Crafting game. The objective of the game is to get planks by chopping trees (which changes the plank's attribute from `absent` to `in inventory`, and the tree's attribute to from `in world` to `absent` as the resource is depleted), then craft wooden stairs using the plank (which depletes the plank resources to `absent` and creates wooden stairs `in inventory`). Comparing the true state trajectory with the imagined plan at t=0, we notice that the imagined future is incorrect: modelling only single item changes do not capture the fact that resources are depleted, but only new resources are created. Nevertheless, at t=1, the model observes the new abstract state and can therefore re-plan with the most up-to-date information which corrects for previously incorrect predictions. A case like this happens in practice within the 2D Crafting games.

**An adversarial failure case**    Here we artificially design an adversarial setting which is not encountered in our evaluation environments, but would in theory result in sub-optimal plans. With reference to Figure 10a, we construct an environment whose goal is to construct *both* a wooden stair *and* a wooden door. Both require a single unit of wooden plank to craft. The attribute here is the quantity of the item in the inventory. Suppose further that once we start crafting, we can no longer collect additional wooden planks.

The correct sequence of behaviour would be to first collect another unit of wood, before crafting a wooden stairs and a wooden door. Planning with a discriminative single item change model would result in a sub-optimal plan, as it fails to consider that crafting a wooden stair result in the wooden plank being consumed, and therefore cannot subsequently craft the wooden door (Fig.10a, plan at t=0).

Note that this failure case is a limitation of doing discriminative modelling with single item changes (Section 3.1), not the Ab-MDP framework. Doing generative modelling (considering all possible $X_{t+1} \in \mathcal{X}$ given $X_t, b_t$) would be able to learn the correct relationship—at the cost of data inefficiency (Section 4.4.1). Nevertheless, we discuss how to fix it below with discriminative modelling.

**Discriminative modelling of multi-object changes**    One simple way of generating plans with multi-item dependencies over time is to consider multi-item behaviours (Section B.1.5). With reference to Figure 10b, we notice that simply modelling two item changes can perfectly capture the dynamics within this task and generate the optimal plan. This comes at a cost of a greater number of behaviours to consider at each step of planning (E.g. for $N = 3$ items, $m = 3$ attributes, there are 9 behaviours to consider for single-item changes, and 27 behaviours to consider for two-items changes).

Generally speaking, with a small change in the representation of the problem, we can plan to generate the optimal sequence of abstract behaviours. This can also happen through changing the abstract mapping—e.g. by modelling items along with the resources needed to create them. Overall, we view

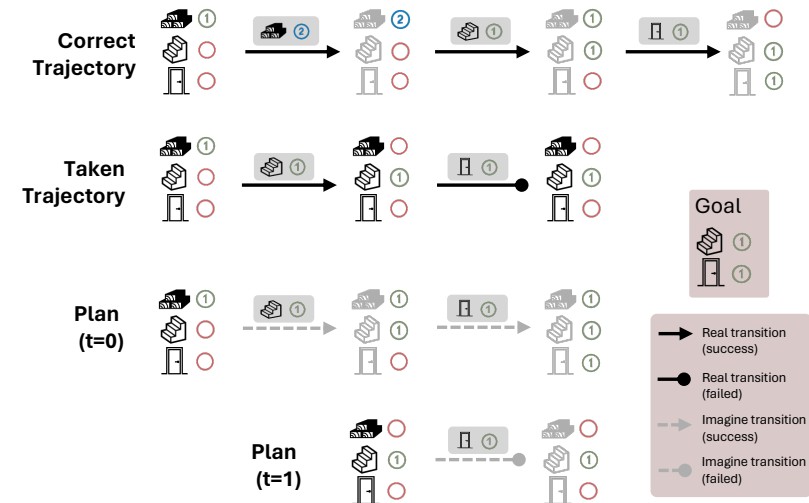

(a) Planning with single item-attribute changes fails in this adversarial task.

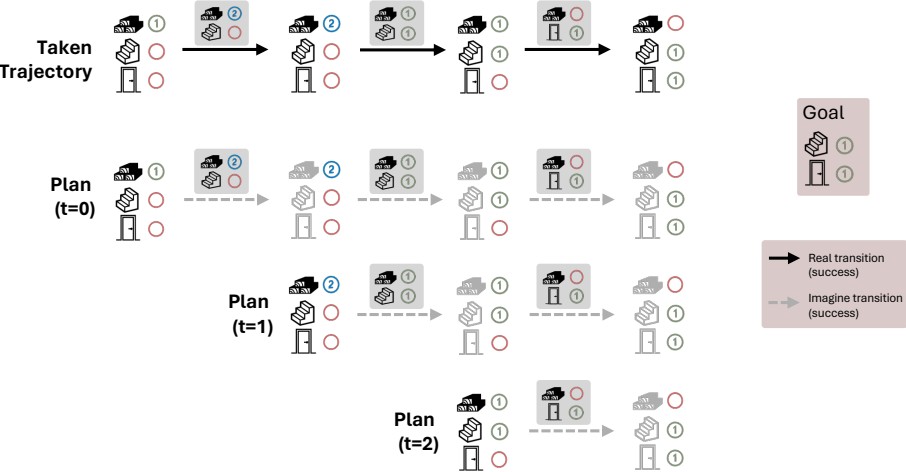

(b) Planning with double item-attribute changes models attribute dynamics perfectly.

Figure 10: Single and double item-attribute change planning

this as a problem of constructing the right abstraction for the problem which, although important, is not the main focus of this work and nor a fundamental issue with our modelling choice.

## C  MEAD: EXTENDED METHODS

Here we provide more details about our methods

### C.1  MODEL FITTING

Given a dataset of (abstract) trajectories collected via the exploration strategy outlined in Section 3.2, $\mathcal{D} = \{(X_1, b_1, X_2, b_2, ...)_i\}_{i=1,...,n}$, we can estimate the empirical success probability $q$ of all observed item-attribute pairs by counting their occurrences:

$$q(X, b) \approx \rho(X, b) = \frac{\text{Number of } (X_t, b_t, X_{t+1}) \text{ transitions where } (\alpha^{(i)}, \xi') \in X_{t+1} + \epsilon}{\text{Number of } (X_t, b_t) + 2\epsilon}, \quad (9)$$

where $\epsilon$ is a small smoothing constant (usually 1e-3). This can be efficiently implemented as a hash map, via hashing $(X_t, b_t)$ as keys, and the number of successful and total transitions (starting from $(X_t, b_t)$) as values.

To train our discriminative model $f_\theta : \mathcal{X} \times \mathcal{B} \to [0, 1]$, we minimize binary cross entropy loss with $\rho$ as the target,

$$\theta^* = \arg\min_\theta \mathbb{E}\left[\rho(X, b) \cdot \log\left(f_\theta(X, b)\right) + (1 - \rho(X, b)) \cdot \log\left(1 - f_\theta(x, b)\right)\right], \qquad (10)$$

where the expectation is estimated by uniformly sampling unique $(X, b)$ pairs from dataset.

Appendix C.2 contains architectural details for $f_\theta$. All in all, we found that our sampling method and loss function lead to a stable, simple-to-optimize objective. Moreover, it is possible to directly use $\rho$ to do planning, as is done in Zhang et al. (2018). However, we show in Section F.1.3 that it is more efficient to use our parametric modelling approach as it generalizes better to new objects.

---

**Algorithm 1** Example Data Collection and Training loop for MEAD

---

**Require:** Model $f_\theta$
  **repeat**
    Collect $n$ frames of data in environment using model $f_\theta$ to plan to explore
    Add data to dataset $\mathcal{D}$
    Reset model weights $\theta$
    **while** accuracy requirement not met **do**
      **if** local minima **then**
        Reset model weights $\theta$
      **end if**
      Sample from $\mathcal{D}$ and minimize Equation 10, for $m$ optimizer steps
    **end while**
  **until** time runs out

---

We provide the pseudocode for training MEAD in Algorithm 1. The accuracy requirement is a running average binary classification accuracy (typically set to 0.95). The local minima is evaluated by checking if the difference is accuracy is close to zero (taken as a exponentially smoothed average to be insensitive to stochasticity). We find in Figure 28 that the most important setting is to run enough gradient updates between data collection steps—the precise setting of checking for accuracy and local minima does not matter as much.

## C.2   MODEL ARCHITECTURE

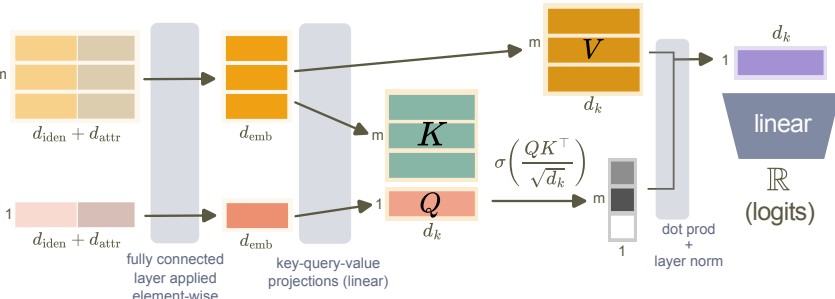

Figure 11: Model Architecture

We use a small key-query attention architecture that takes in sets of vectors to produce a binary prediction (Figure 11). Each item $x^{(i)} = (\alpha^{(i)}, \xi^{(i)})$ is represented by its item identity and item attribute embeddings. An Ab-MDP state containing a set of $m$ items is represented as a $m \times (d_{\text{iden}} + d_{\text{attr}})$ matrix. The behaviour $b = (\alpha^{(i)}, \xi')$ describes an item identity and the *new* attribute to try to change it to, and is a $d_{\text{iden}} + d_{\text{attr}}$ dimensional vector. The set of items and the behaviour is passed

element-wise through a small MLP (specifically a 2 layer ReLU network with 128 units in its hidden layer) to produce a set of $m \times d_{\text{emb}}$ item embeddings, and a $d_{\text{emb}}$-dimensional behaviour embedding ($d_{\text{emb}} = 128$). The item embeddings are linearly projected into key and value matrices $K$ and $V$ both with shape $m \times d_k$, and the behaviour embedding is linearly projected into a $1 \times d_k$ query vector $Q$ ($d_k = 256$). We compute the scaled dot product attention (Vaswani, 2017):

$$\text{Attention}(Q, K, V) = \text{softmax}\left(\frac{QK^\top}{\sqrt{d_k}}V\right) . \tag{11}$$

The output of the attention operation is an $1 \times d_k$ vector. Finally, it is linearly projected to a scalar, representing the un-normalized logit for binary prediction.

### C.3 EXPLORATION

We use a reward function that encourages indefinitely exploring all states and behaviours,

$$r_{\text{intr}}(X, b) = \sqrt{\frac{T}{\text{N}(X, b) + \epsilon T}} , \tag{12}$$

where $\text{N}(X, b)$ is a count of the number of times $(X, b)$ is visited, $T$ is the total number of abstract time-steps, and $\epsilon = 0.001$ is a smoothing constant.

Intuitively, a state with many visits (large $\text{N}(X, b)$) will have lower reward, while $T$ prevents the intrinsic reward from shrinking to zero (so the agent explores indefinitely). Indeed, in the bandits setting, this reward function is maximized when all states are visited uniformly (Zhang et al., 2018). We emphasize that *all* of our model training is done with this intrinsic reward, with no task-specific information injected at training time.

We use MCTS to find high (intrinsic) reward states. MCTS iteratively expands a search tree and balances between exploring less-visited states (using an upper confidence tree) and exploiting rewarding states.[6] In our case, MCTS generates child nodes of abstract state $X$ by proposing behaviours $b$ and checking if $f_\theta(X, b) > 0.5$ (Figure 3a). We also introduce a small degree of randomness so all behaviours have a small chance of being selected.

We modify the MCTS algorithm to return the the maximum $r_{\text{intr}}(X, b)$ encountered along each path during the back-propagation stage and set a fix expansion depth without a simulation stage from the leaf node. This is done to prevent loops and encourage the agent to go toward frontier states with the least amount of visitations.

## D ENVIRONMENTS

### D.1 2D CRAFTING

We adopt three crafting environments proposed in Chen et al. (2020), which contain Minecraft-like crafting logic. The agent needs to traverse a grid world, collect resources, and build increasingly more complex items by collecting and crafting ingredients (e.g. getting an axe, chopping tree to get wood, before being able to make wooden planks and furniture that depend on planks). Primitive actions consist of movement in the four cardinal directions (N, E, S, W), toggle (of a switch to open doors), grab (e.g. pickaxe), mine (e.g. ores), craft (at crafting bench of specific items). The low level observation space consist of a dictionary containing: (i) an (5, 5, 300) matrix describing a $5 \times 5$ world with each state being described by an 300-dim word embedding, (ii) an 300-dim inventory embedding describing things in the inventory, and (iii) a 300-dim goal embedding describing in words the task to complete. We describe each environment's crafting sequences required to finish each task in Table 2.

---

[6]This exploration is purely within the *model's imagination*, rather than the real environment. Balancing exploration and exploitation is still necessary as exhaustive search in the abstract space is highly inefficient / intractable.



Figure 12: Example episode in an example 2D Crafting environment, figure taken from Chen et al. (2020). To solve this task, the agent needs to pickup the pickaxe, mine the cobblestone stash to get cobblestone, then use the acquired cobblestone to craft cobblestone stairs. The items must be acquired in that order as each step requires having the item from the previous step (with an additional obstacle of being able to cross the door only if it has a key).

| Environment Name | Crafting Task Description |
|---|---|
| 2 Steps | Pickup pickaxe, mine gold ore |
| 3 Steps | Pickup pickaxe, mine diamond ore, craft diamond boots |
| 4 Steps | Pickup axe, chop tree to get wood, craft wooden plank from wood, then craft either wooden stairs or wooden door from plank |

Table 2: 2D Crafting environments.

**Abstraction States**   We build a simple abstraction to map from the low level observation to an abstract representation. We take as items all objects in the world, in the inventory, and potentially craft-able objects. We assign each item one of the the attributes in Table 3.

We take each item identity as its 300-dim word embedding provided from the environment, and each attribute as a one-hot embedding, repeated to also be 300-dim. Each item in the item set is described as a 600-dim vector.

| Attribute | Description |
|---|---|
| IN_WORLD | An item is in the world but not in the inventory. |
| IN_INVENTORY | An item is in the player's inventory. |
| ABSENT | An item is neither in the inventory or in the world. |

Table 3: 2D Crafting attributes.

## D.2 MINIHACK

We also use the MiniHack environment (Samvelyan et al., 2021), which is built on the Nethack Learning Environment (Küttler et al., 2020). We use five hard-exploration games from Henaff et al. (2022) requiring interaction with different types of objects. The base game has a multi-modal observation space involving image, text, inventory, etc. Solving the game from the base observation is extremely difficult since each environment has multiple variants of an object (e.g. Freeze-Wand has 27 different types of wands), and randomly generated object locations. Indeed, Henaff et al. (2022) found that performant RL methods such as IMPALA (Espeholt et al., 2018; Küttler et al., 2019) and all tested global exploration methods (ICM (Pathak et al., 2017) and RND (Burda et al., 2019)) fail to solve these environments.

The base environment contains an dictionary observation space, containing a $21 \times 79$ grid of glyphs (integer denoting one of 5976 nethack glyph types), an 27-dimensional vector denoting agent's stats, a 256 dim vector for message, and another vector denoting agent's inventory. The environments are:

- `"MiniHack-Levitate-Boots-Restricted-v0"`
- `"MiniHack-Levitate-Potion-Restricted-v0"`
- `"MiniHack-Freeze-Horn-Restricted-v0"`

- `"MiniHack-Freeze-Wand-Restricted-v0"`
- `"MiniHack-Freeze-Random-Restricted-v0"`

Generically, the Nethack Learning Environment contains over 100 actions; whereas the "restricted" set of games reduce this space. The possible primitive actions include the 8 cardinal actions (`N`, `E`, `S`, `W`, `NE`, `SE`, `SW`, `NW`), `PICKUP`, `ZAP`, `WEAR`, `APPLY`, `QUAFF`, `PUTON`, `FIRE`, `READ`, and `RUSH`. It may be even less dependeing on the specific environment.

**Abstract States**  We hand-construct a map that maps from the multi-modal observations to a set of (object identities, object attribute) vectors. We do this by taking the `glyph` and `inventory` observation modalities and filtering out all glyphs containing blank spaces, floor, and walls. To get the **object attributes**, We take the remaining glyphs and mark them as being "in world" if they are in `glyph`, and "in inventory" if they were from the `inventory` modality. Then, if the agent object was beside an object and took a primitive action to move towards it, we mark the object as "standing on". If we observe a levitating message (e.g. "up, up and away" from the `message` modality, we set the agent object to have "levitate" attribute. Alternatively, if the low level episode terminates along with the last observation being either a levitate message or a "freeze spell" message (e.g. "the frozen bolt bounces"), then we set the corresponding levitate or freeze object to have "used" status. Finally, if an object disappears from the low level observation that is not being stood on, we set it to have "removed" attribute. We summarize this in Table 4.

| Attribute Name | Meaning | How to Infer |
|---|---|---|
| in world | The object is in the world | Object is in the glyph observation |
| standing on | Object in world, and the player is standing on it | Object was in the glyph observation and message says "you see here..." |
| in inventory | Object in inventory | Object in the inventory observation |
| levitating | Agent is levitating | Message says a levitate message |
| used | An object is used | A freeze or levitate message appears, and episode terminates |
| removed | Object disappears from world | The object is not longer present |

Table 4: Object Attributes and how to infer them.

As for **object identities**, they are simply the glyph number given to them in MiniHack (e.g. 329 for the player agent).

**Embeddings**  We then embed the **object identities** and attributes before providing them to the agents. For object identities, we take each object's glyph number, and map it to the NetHack text description for that glyph (which we can extract from the `screen_descriptions` modality in the low level observation). For each glyph's text description (which is a short sentence), we pre-process the sentence to remove stop words, then take each word's GloVe (Pennington et al., 2014) embedding and averaging over the sentence. We use the 300-dimensional Wikipedia 2014 + Gigaword 5 GloVe word vector embeddings from https://nlp.stanford.edu/projects/glove/ (6B tokens, 400k vocab, uncased). The 300-dim vectors were chosen as we found them to have naturally good separation between object types in preliminary analysis.

We embed the **object attributes** as 300-dimensional one-hot vectors (i.e. one-hot between six categories and repeated to fill 300 dimension). This is done to make sure the input space is equally represented by the *identity* and *attribute* embeddings.

### D.2.1 SANDBOX ENVIRONMENT

We design a sandbox environment where the agent can observe and interact with all items it may encounter in one of the five evaluation environments. In the sandbox, each of these objects has an 0.5 chance (independently) of spawning at a random location: levitate boots, levitate rings, levitate potions, freeze horn, freeze wands. Further there is a 0.5 change that a second of the same object type will spawn. The agent can interact with any of the objects for up to 250 low level frames, or the episode terminates once the agent sets any of the object's attribute to "used". Figure 14c shows an instantiation of an episode.

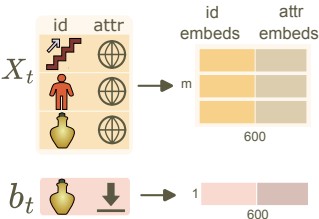

Figure 13: Embedding

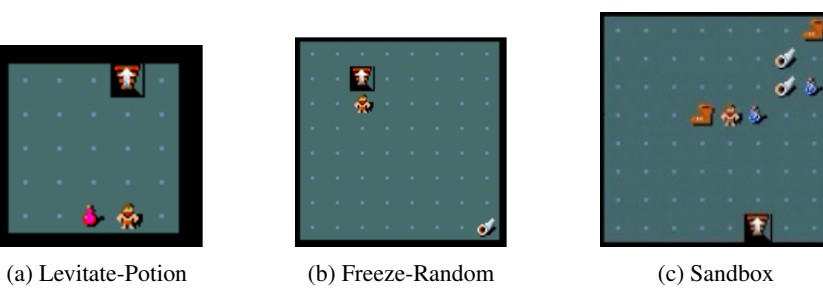

(a) Levitate-Potion        (b) Freeze-Random        (c) Sandbox

Figure 14: Example Minihack environments. Only a cropped version of the screen is rendered for presentation clarity. The full observation space contains a larger screen, agent's inventory, in-game stats, and in-game messages (not shown here).

### D.2.2 LOW LEVEL POLICY ERRORS

The learned and hand-defined low level policies are not perfect. We see that depending on the initial abstract state, the low level policy succeeds on average around 90% of the time, but can be as low as 10% for particular states (Figure 15).

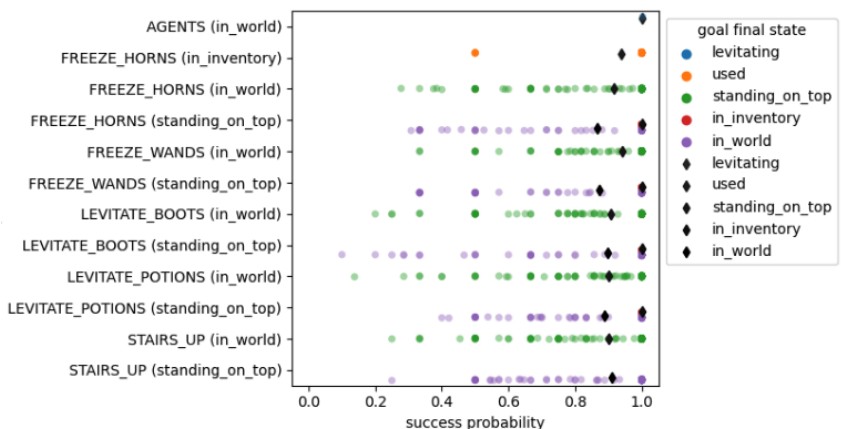

Figure 15: Low-Level policy success rates for an environment

Some common failure cases for these low-level policies are:

1. To use an item in the inventory, the policy needs to select which item to use (e.g. to drink a potion the policy needs to first do primitive action "quaff", then primitive action that selects the inventory object to drink, etc.). When there are multiple objects in the inventory it just picks a random object, meaning it might not pick the object specified by the behaviour

2. The "go stand on" policy is coded to move in a straight line towards an object. Since we measure a policy "success" whenever the abstract state changes, if the agent walks over

     another object en route, the Ab-MDP will register an abstract state change and deem the transition unsuccessful

3. To successfully cast a freeze spell, the NetHack game needs to return a message saying the spell has bounced off a wall. To our knowledge, this is stochastic—we got as far as figuring out that the spell needs to travel sufficient distance to bounce, but even so we were unable to produce a low level policy that results in spell bounce 100% of the time. This succeeds $\sim 80\%$ of the time

## E    BASELINE METHODS

For baselines, we use an environment wrapper which abstracts away the low level observations into only the abstract level observations and behaviours (as described in Section 2.1). We similarly equip the baseline policies with transformer-based encoders, with input masking in places where the input sequence is longer than the number of objects in the set as well as positional embedding, which we found in preliminary experiments helped with training. For action spaces, the agent generates a discrete integer action which maps onto an object index in the observation set as well as the object attribute to change to.

Specifically, we use Dreamer-v3 as a performant model-based baseline (Hafner et al., 2023), MuZero as another model-based baseline (Schrittwieser et al., 2020; Werner Duvaud, 2019), and PPO as a model free baseline (Schulman et al., 2017; Petrenko et al., 2020). For exploration-reward transfer experiments, we use a disagreement-based model for Dreamer intrinsic reward (Sekar et al., 2020), as well as using the same count-based reward to train all models.

Finally, we also compare against other methods that uses a structured MDP, such as the attribute planner in Zhang et al. (2018), and NCS in Chang et al. (2023). These results can be found in Section F.1.3.

We outline the main differences between the different model-learning methods in Table 5.

| | State Representation | Function Approx | Planning |
|---|---|---|---|
| Dreamer-V3 (Hafner et al., 2023) | Feature vector $\phi$ | Neural Net (generative) | Model free (in imagination) |
| MuZero (Schrittwieser et al., 2020) | Feature vector $\phi$ | Neural Net (generative) | MCTS |
| DAC-MDP (Shrestha et al., 2020) | Feature vector $\phi$ | k-NN | Value iteration (on k-NN "cores") |
| Attribute Planner (Zhang et al., 2018) | Tabular | No approximation | Dijkstra |
| DLP-EIT (Haramati et al., 2024) | Sets of vectors | Neural Net (generative) | Model free |
| NCS (Chang et al., 2023) | Sets of $(\alpha, \xi)$ vectors | K-means (on just attribute vectors $\xi$) | Greedy |
| MEAD | Sets of $(\alpha, \xi)$ vectors | Neural net (discriminative) | Dijkstra |
| Ablation | Sets of $(\alpha, \xi)$ vectors | K-means (on just attribute vectors $\xi$) | Dijkstras |

Table 5: Model-based methods

## F    EXTENDED RESULTS

### F.1    EXTENDED MAIN RESULTS

All runs, extended and main, are run with a minimum of 3 independent seeds. We outline the specific number of seeds for each method and each main experiment in Table 6.

| Experiment Group | Method | N Seeds |
|---|---|---|
| Learning from scratch, 2D crafting games (Figure 4a) | MEAD
Dreamer-V3
PPO | 4
3
3 |
| Learning from scratch, Minihack skill games (Figure 4b) | MEAD
MuZero
Dreamer-V3
PPO | 5
3
4
3 |
| Transfer from Sandbox pre-training (Figure 5a) | MEAD
Dreamer-V3 (count / disagreement / reward) | 3
3 / 3 / 3 |
| Transfer from Freeze-Random pretraining (Figure 5b) | MEAD
Dreamer-V3 (count / disagreement / reward) | 3
3 / 3 / 3 |
| Compositional Environment (Figure 5c) | MEAD (from scratch / pretrained)
Dreamer-V3 (from scratch / pretrained) | 3 / 3
3 / 3 |

Table 6: Number of seeds used in our main results.

To get the performance of the various algorithms in the low level MDP as we plotted them in Figure 4b, we extracted them from Figure 13 in Henaff et al. (2022), using the WebPlotDigitizer app (https://apps.automeris.io/wpd/).

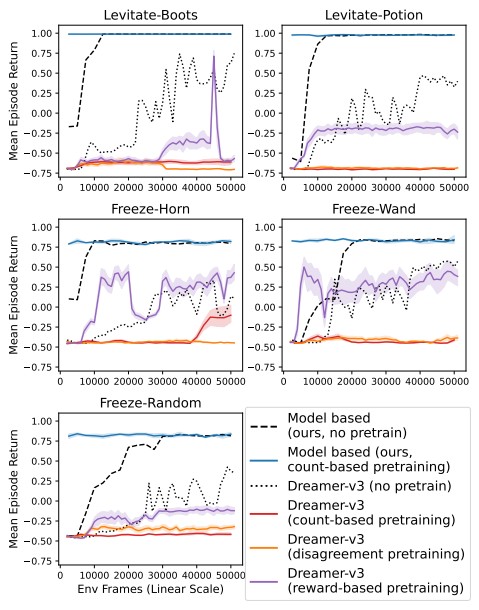

(a) Sandbox transfer performance. Episode return from fine-tuning model weights from pre-training in a sandbox environment containing multiple objects instances and types.

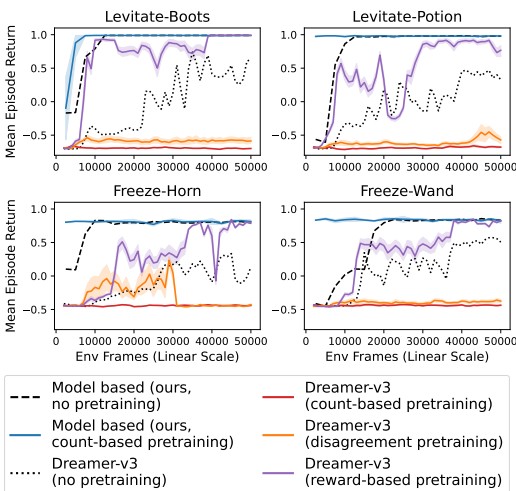

(b) Transfer to new object types. Pretrained in `Freeze Random` environment where `Freeze Horn` and `Freeze Wand` objects appear with 0.5 probabilities, fine-tuning on remaining 4 environments. (Top row): fine-tuning performance where new object type is fully unseen, (bottom row): fine-tuning performance for seen objects but appearing in different frequencies.

Figure 16: Full transfer experiments

### F.1.1 PPO ALL RUNS

We plot all PPO runs with different pre-training settings in Figure 18. Pre-training resulted in only negative transfer as compared to no pre-training.

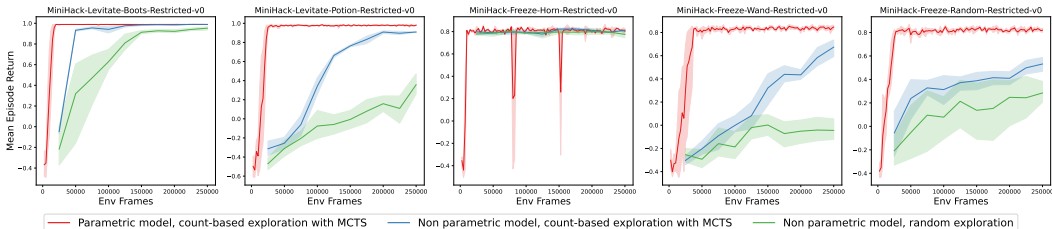

Figure 17: Training from scratch with different model types

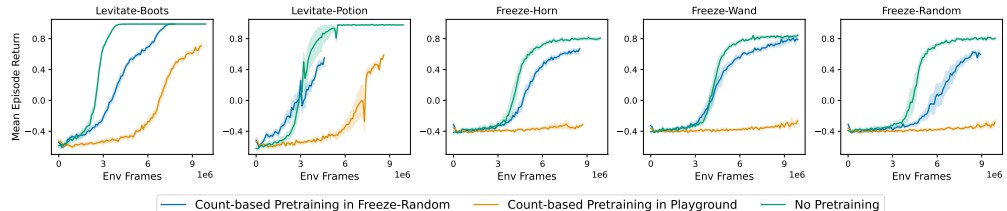

Figure 18: PPO all runs.

### F.1.2 DREAMER

We show extended transfer results for Dreamer. We see in Figures 19 and 20 that when trained for much longer, Dreamer with both disagreement-based and count-based pre-training does start to get more rewards. However, we also observe the reward curves with pre-training is much worse than the reward curves without pre-training (black dotted lines), indicating negative transfer across environment and object types for these methods.

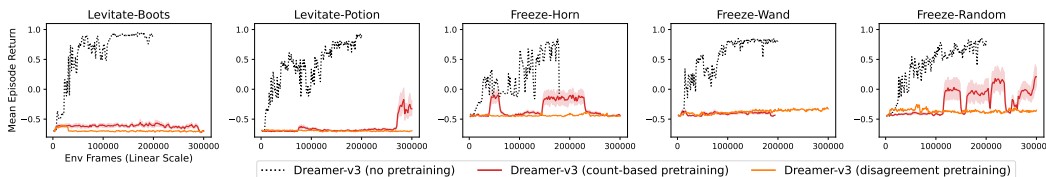

Figure 19: Longer Dreamer pretraining runs in the sandbox environment to show eval reward does increase with more samples, albeit pretraining seems to result in negative transfer.

### F.1.3 OTHER MODEL LEARNING APPROACHES

We plot comprehensively the non-parametric model results in Figure 21. This model uses a table to keep track of transition successful probabilities between abstract states, as used by the attribute planner in Zhang et al. (2018). We show direct comparison between methods in Figure 17.

We further compare against an approach using K-means clustering on the attributes, and labeling edges (between the k-means "cores") with the last observed action that successfully led to a transition. This way of model-learning along with greedy planning was used by the NCS (neural constraint satisfaction) method of Chang et al. (2023). Specifically NCS uses k-means clustering with greedy planning, which we found to not work for any of our environments. A hybrid variant using k-means clustering with Dijkstra planning works in MiniHack (albeit less well than MEAD), and fails completely in 2D crafting.

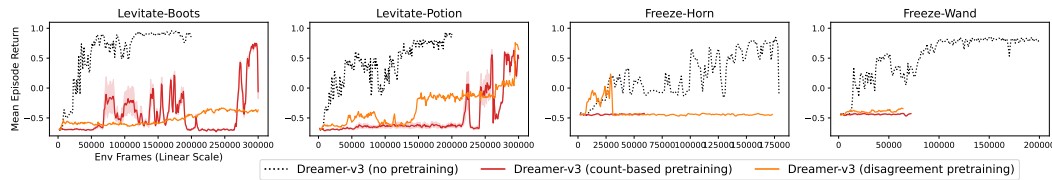

Figure 20: Longer Dreamer pretraining runs in the `Freeze Random` environment to show eval reward does increase with more samples, ableit pretraining seems to result in negative transfer.

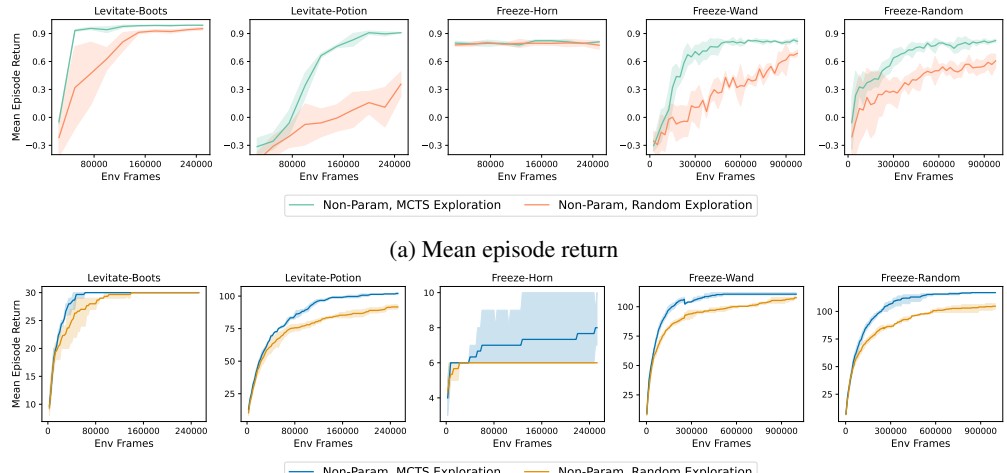

(a) Mean episode return

(b) Number of unique (Object, Attribute, New Attribute) transitions discovered

Figure 21: Non-parametric agent. Notice the difference in X-axis scale for Freeze-Wand and Freeze-Random environments.

## F.2   LOW LEVEL LEARNING

### F.2.1   LEARNING LOW LEVEL POLICIES

We can reinforcement learn low level behaviours. From some $X_t$, we propose behaviour $b = (\alpha^{(i)}, \xi')$ and execute $\pi_b$ until an abstract transition. If the resulting $X_{t+1}$ contains item $(\alpha^{(i)}, \xi')$, we reward $\pi_b$ with a reward of +1 (else -1).

While this can be implemented in many ways, in practice we use a goal conditioned RL set-up, where a single model free policy network takes in both the low level state and the behaviour embedding to produce a primitive action: $\pi : \mathcal{S} \times \mathcal{B} \to \mathcal{A}$. The low level state is the environment's raw observations while the behaviour embedding is the same as the one given to the abstract level model. This is implemented as a wrapper over the original environment which provides the policy with a +1 reward if $(\alpha^{(i)}, \xi') \in X_{t+1}$, and -1 otherwise. We use APPO as the policy optimization algorithm (Petrenko et al., 2020). The point of this is merely to show that we can learn the set of behaviours neurally and building an Ab-MDP with them still works.

The full MiniHack low level policy learning result is shown in Figure 24a. The performance of MEAD in an Ab-MDP with learned behaviours is shown in Figure 24b. Note that the reported 2D Crafting results in main text Section 4.1 already use neurally learned behaviours.

### F.2.2   LEARNING OBJECT-CENTRIC MAP

We train an object centric map which takes the low level observation and predict the abstract state: $\hat{X}_t = M_\theta(S_{t-1}, A_t, S_t)$. The encoder map $M_\theta$ uses a convolutional encoder for MiniHack low level observations (specifically, the low level encoder from Henaff et al. (2022)), and learned embeddings

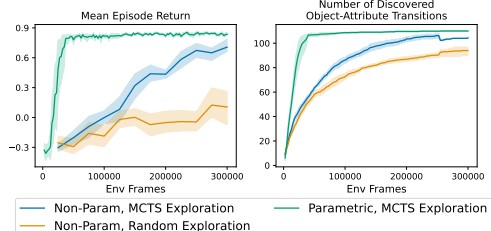
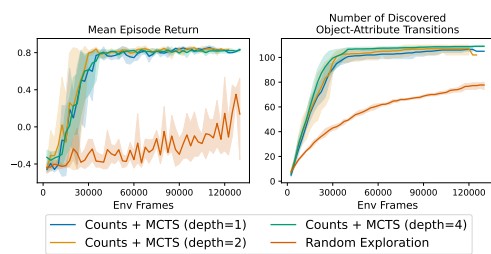
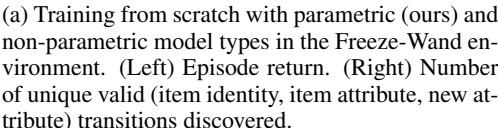

(a) Training from scratch with parametric (ours) and non-parametric model types in the Freeze-Wand environment. (Left) Episode return. (Right) Number of unique valid (item identity, item attribute, new attribute) transitions discovered.

(b) Effect of count-based intrinsic reward and MCTS in the Freeze-Wand environment. (Left) Episode return. (Right) Number of unique valid (item identity, item attribute, new attribute) transitions discovered by the agent.

Figure 22: Model ablations

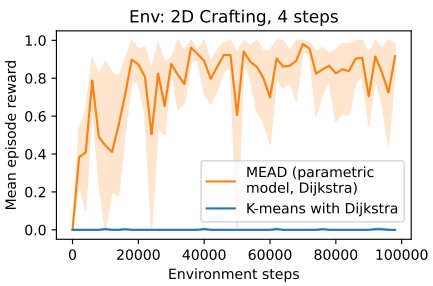
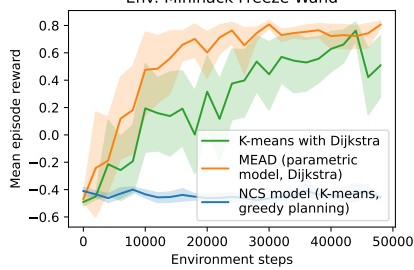

(a) Training from scratch in 2D crafting env.

(b) Training from scratch in MiniHack.

Figure 23: Performance of the k-means clustering on attributes method as used in NCS (Chang et al., 2023). NCS specifically uses k-means clustering with greedy planning, although we also test a variant using k-means clustering with Dijkstra planning. Legends specify the function approximation and planning methods used.

for each discrete actions. After concatenating all encoded inputs, the latent is projected through two 2-layer feedforward nets, one producing a categorical distribution over item identities, and another a categorical distribution over item attributes. The output of this map are two tensors each with shape (item set size $\times$ num categories). We can train this via a categorical cross-entropy loss to predict the correct identity and attribute categories for all items.

We collect a dataset of 100k transitions using a random policy in the MiniHack Freeze-Horn environment, and use the hand-crafted mapping to provide ground truth abstract state labels. We train using Adams optimizer with batch size 4096 and learning rate 3e-4.

We investigate the effect of training with different sub-sets of the data on both the encoder accuracy and the subsequent planning performance in Figure 25.

## F.3 Extended Ablation Results

### F.3.1 Generative and Discriminative Modelling

We design an experiment where we isolate the difference made by the generative vs. discriminative modelling, which we can view as a constraint over the support of the predicted distribution for $X_{t+1}$.

We collect a dataset using a hand-crafted expert policy in MiniHack's Freeze-Horn environment, with a degree of action noise added to increase coverage. This is done to remove the exploration problem: the model only needs to learn to fit the trajectories well and it should contain within its world model the correct transitions required to solve the task. We fit two models:

- A discriminative model, which learns to predict success probability (Equation 1) as described by the main text.

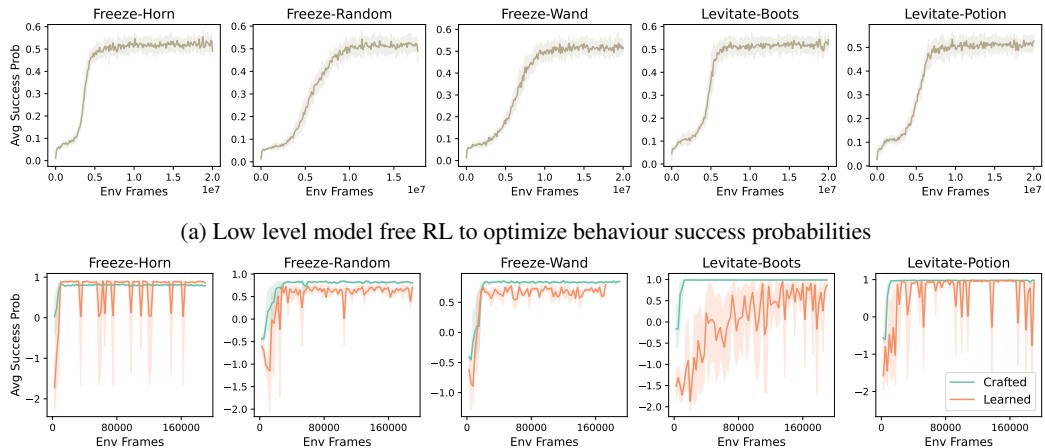

(a) Low level model free RL to optimize behaviour success probabilities

(b) MEAD performance in Ab-MDP with learned vs. crafted behaviours

Figure 24: Learning low level behaviours within MiniHack

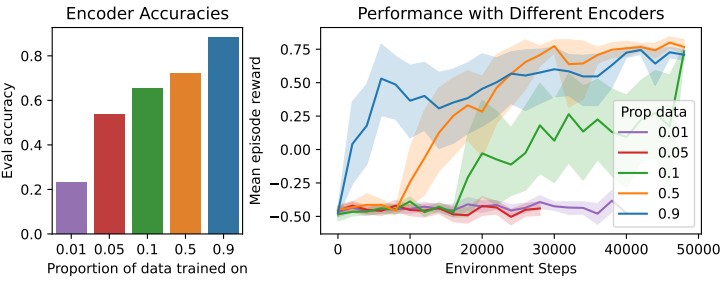

Figure 25: Effect of training on different proportions of the 100k dataset on encoder accuracy (left) and MEAD performance (right) in `Minihack-Freeze-Horn`.

- A generative model, which learns to predict the distribution over $\mathcal{X}$.

The two models use the same architecture to encode the previous abstract state and behaviours. It differs only in its output head: the discriminative model outputs a binary prediction, while the generative model outputs a categorical prediction for each item. We observe in Figure 26 that both model classes optimize the loss.

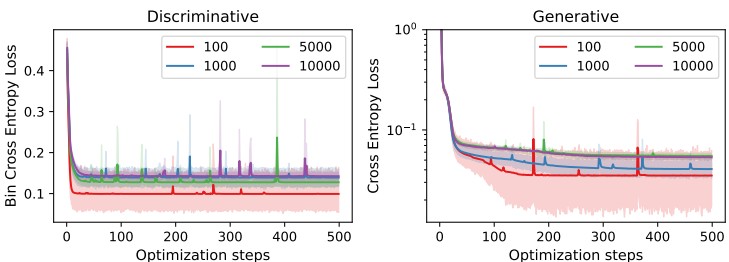

Figure 26: Loss curves for training generative and discriminative mdoels on a fixed expert dataset. Colors denote different number of dataset transition size.

To do planning, we run Dijkstra, with the edge weight as the negative log of $\Pr(X'|X, b, \theta)$ (where $\theta$ denotes the generative or discriminative model). There are two ways we can generate the neighbours for the generative model: either by proposing the modal next state, i.e. $X' = \arg\max_{X'} \Pr(X'|X, b, \theta)$, or by expected behaviour change $X' = \Delta(X, b)$. Using ex-

pected behaviour change does not work in planning (the model never proposes behaviour sequences that lead to solving the task), thus we report results by using the modal next state. The results are reported in main text Figure 7, noting the error bars denote standard error of mean.

### F.3.2 MODEL FITTING

We found that running sufficient number of model fitting steps in between data collection steps was important. Fully re-setting the weights before data fitting is also helpful.

### F.3.3 GLOVE SEMANTIC EMBEDDING

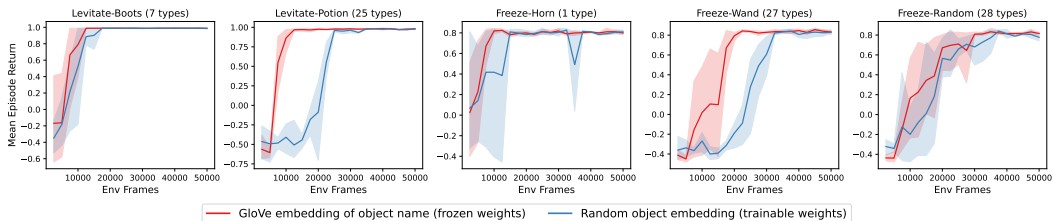

Figure 27: Training from scratch with frozen GloVe embeddings, or trainable randomly initialized weights.

We investigate the impact of having the object identity embeddings be semantically meaningful versus random one-hot embedding in MiniHack. By default, all experiments in the main text take as input object identity embedding using GloVe Pennington et al. (2014) embedding, which are kept frozen throughout training. We ablate this by initializing the object identity embedding matrix to be randomly initialized from a standard Gaussian, $\mathcal{N}(0, 1)$, and allow them to be updated by gradients.

Figure 27 shows the results in all five games. We observe that having word embedding helps a bit in environment where there are many types of the same objects (i.e. Levitate-Potion and Freeze-Wand). This is unsurprising as the text embedding between objects would be rather similar (e.g. "wooden wand" and "metal wand" would have similar averaged embedding). Abeit they still performant overall. This hints at the abstraction being useful mainly due to its (object identity, object attribute) structure.

### F.4 MODEL UPDATES

We show effects of number of model update steps during data collection in Figure 28, 29,&31.

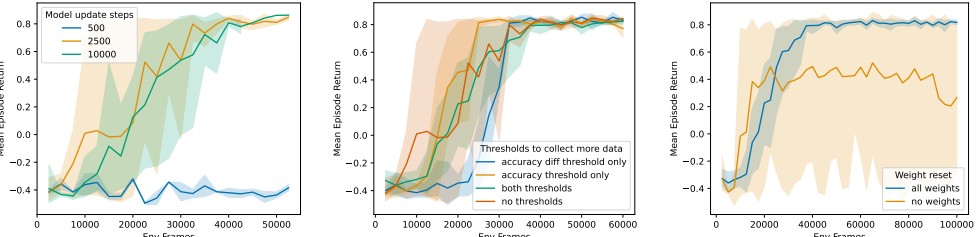

(a) Effect of model update steps between data collection steps.

(b) Effect of model threshold logic prior to being allowed to collect more data

(c) Weight reset strategies before fitting on new data

Figure 28: Model training hyperparameters training in `Freeze Wand` environment.

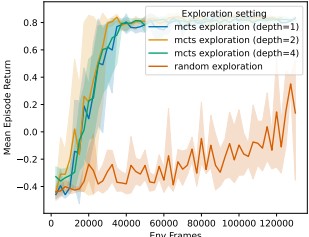
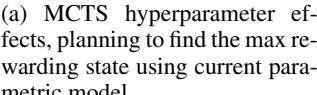

(a) MCTS hyperparameter effects, planning to find the max rewarding state using current parametric model.

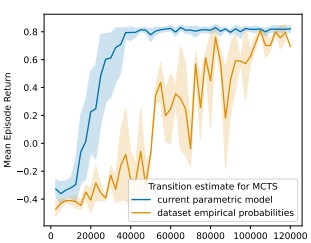

(b) Using the empirical dataset probability for MCTS planning (i.e., a non-parametric estimate)

Figure 29: MCTS exploration hyperparameters in the `Freeze Wand` environment.

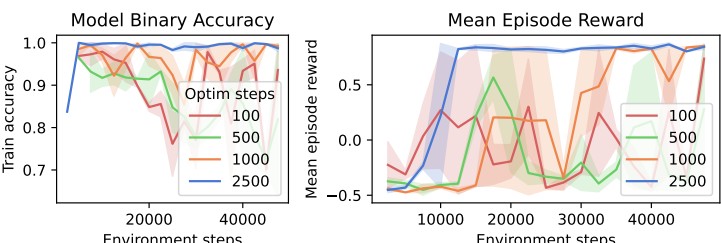

Figure 30: Ablating just the effect of model optimization steps in `MiniHack-Freeze-Wand` env. Model weights are reset after each data collection step (of 2.5k frames), then trained for the specified number of steps before the next data collection step. With sufficient optimization steps, high model accuracy and planning performance is achieved.

### F.5 MODEL EMBEDDING

We investigate the vector representation of different (object identity, object attribute) pairs in our model at the start of training and in the end. Results are shown in Figure 32. We observe vector cluster mainly along object attributes in the start of training (due to the orthogonal object embedding initialization). After training, vectors cluster by object attributes *in an identity-dependent way*. For instance, the `STAIRS_UP` object in Figure 32a are not clustered by attributes along the first two principle components in the Freeze Random environment, likely due to their usage being very different as compared to the other usable objects. We also observe separation by object-identities inside each of the object attribute clusters after training, but not before.

Again referring to Figure 32 (a), we notice that unseen objects (not seen in the `Freeze Random` environment—here it is the "levitate" objects) largely fall into clusters by attributes.

### F.6 INTERNAL WORLD GRAPHS

We can also show the model's internal world model in a highly interpretable way: through running a graph search algorithm (e.g. breadth first search) along transitions the model predicts is high probability, and reconstructing the model's internal graph.

We do this for two environments, showing only high probability transitions, and disallow visualization of loops in the graphs. The world model graph for Freeze-Random is in Figure 33. Note this is the same model which we illustrated in brief in main text Figure 3b. The world model graph for a 200k frames checkpoint in the Levitate-then-Freeze environment is in Figure 34, which is much more complex even with only the high probability edges.

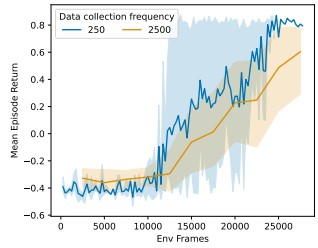 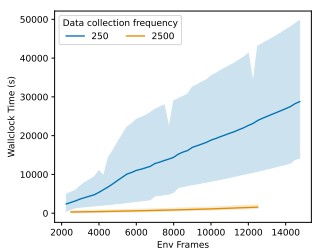

(a) Eval performance as function of data collection

(b) Wallclock time for different data collection frequencies

Figure 31: Data collection frequency in the `Freeze Wand` environment.

# G  TRAINING DETAILS AND HYPERPARAMETERS

## G.1  OUR MODEL

We describe our default parameters in Table 7.

| Hyperparameter | | Value |
|---|---|---|
| | Optimizer | Adam |
| | Learning rate | 1e-4 |
| Optimization | Adam betas | (0.9, 0.999) |
| | Adam eps | 1e-8 |
| | Max batch size | 2048 |
| | Additional frames before training | 2500 |
| Data Collection | Weight reset before training | reset all weights |
| | Minimum optimizer steps before data collection | 2500 |
| | Random goal selection probability | 0.2 |
| MCTS Exploration | Num simulations | 16 |
| | Max search depth | 4 |
| | Probability for valid edge | 0.5 |
| Eval Planning | Dijkstra max iters | 100 |
| | Dijkstra low probability cut-off | 0.1 |

Table 7: Default training parameters for our model

## G.2  DREAMER-V3

We describe the default parameter used to run the Dreamer-v3 baseline. We use a PyTorch implementation of Dreamer: https://github.com/NM512/dreamerv3-torch, which has similar training curves as reported initially in Dreamer-v3. Unless otherwise mentioned, we use the "XL" world model setting with the highest amount of training iteration, which had the best reported sample efficiency in Hafner et al. (2023), and was confirmed by our preliminary experiments (Figure 35). We start with parameters as faithful to the reported ones in Hafner et al. (2023) as possible as it was found to work across a wide range of settings, then sweep over a subset of hyperparameters and report the best ones (which we use) in Table 8.

We perform parameter sweep over a subset and report the best ones (which we use) in Otherwise we keep the parameters as faithful to the reported ones in Hafner et al. (2023) as possible as it was found to work across a wide range of settings.

## G.3  MUZERO

We use the MuZero code base from Werner Duvaud (2019). We sweep over learning rates $\{1 \times 10^{-5}, 1 \times 10^{-4}, 1 \times 10^{-3}\}$, encoding size $\{32, 64, 128, 256, 512\}$, number of simulations $\{30, 90, 100,$

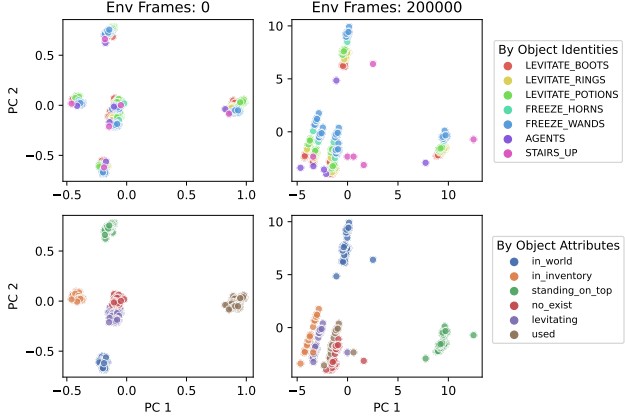

(a) Training in `Freeze Random` environment. Per dimension variance ratio explained at 0 frame: (0.24735658, 0.22067907); at 200k frames: (0.49951893, 0.35679522)

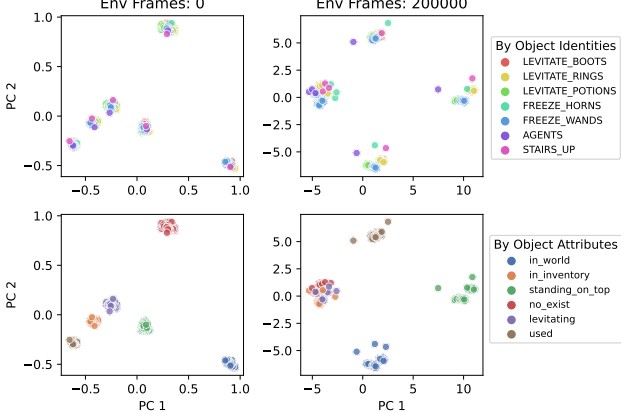

(b) Training in the `Sandbox Playground` environment. Per dimension variance ratio explained at 0 frame: (0.3139481, 0.23430094); at 200k frames: (0.5275301, 0.23016657)

Figure 32: Principle components analysis (PCA) of the (object identity, object attribute) vector representation in the model right before the key-query-value projection layer. PCA projects from 128 dimensions to 2 orthogonal dimensions of maximum variance. We show the representation at two points in training: 0 frames and 200k frames (columns). We label each (object identity, object attribute) vector by its identity (top row) and by its attribute (bottom row) separately.

180}, using prioritized experience replay (or not), batch size {128, 2048}, TD steps {4, 10, 20, 50, 100}, and the delay in self-play simulations (allowing for more training steps in between data collections) of {0, 2, 8}. We use the best hyperparameters and report them in Table 9.

We observe a small effect of different self play delay lengths performing better in different games. We choose a delay of 0 as it performed the best in most games. Nevertheless, we report the performance of different self play delay lengths in Figure 36 for completeness.

### G.4 PPO

We use the Sample-Factory implementation of PPO (Petrenko et al., 2020). We run a parameter sweep over a subset and report the best parameters which we use in Table 10.

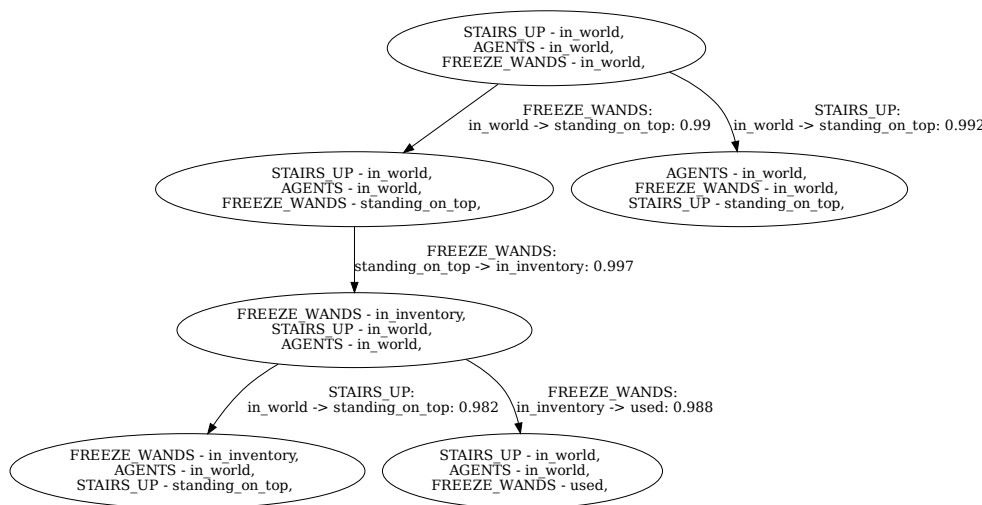

Figure 33: Extracted internal world model with a particular starting state in Freeze-Random by starting traversal from a given initial state. Only the most relevant objects are included for clarity and only transitions with success probability above 0.1 are included. Note that each node has 40-50 edges, although we only visualize the high probability ones.

| Hyperparameter | | Value |
|---|---|---|
| Optimization | Optimizer | Adam |
| | Model lr | 1e-4 |
| | Actor lr | 3e-5 |
| | Critic lr | 3e-5 |
| | Batch size | 32 |
| | Batch length | 64 |
| | Train ratio | 1024 |
| | Replay capacity | $10^6$ |
| Model Units | GRU recurrent units | 4096 |
| | Dense hidden units | 1024 |
| | MLP layers | 5 |
| Planning | Imagination horizon | 15 |
| Exploration (disagreement) | Ensemble size | 10 |

Table 8: Dreamer parameters

| Hyperparameter | Value |
|---|---|
| Learning rate init | $1 \times 10^{-4}$ |
| Encoding size | 512 |
| Batch size | 2048 |
| Num simulations | 100 |
| TD steps | 20 |
| Prioritized experience replay | False |

Table 9: MuZero parameters

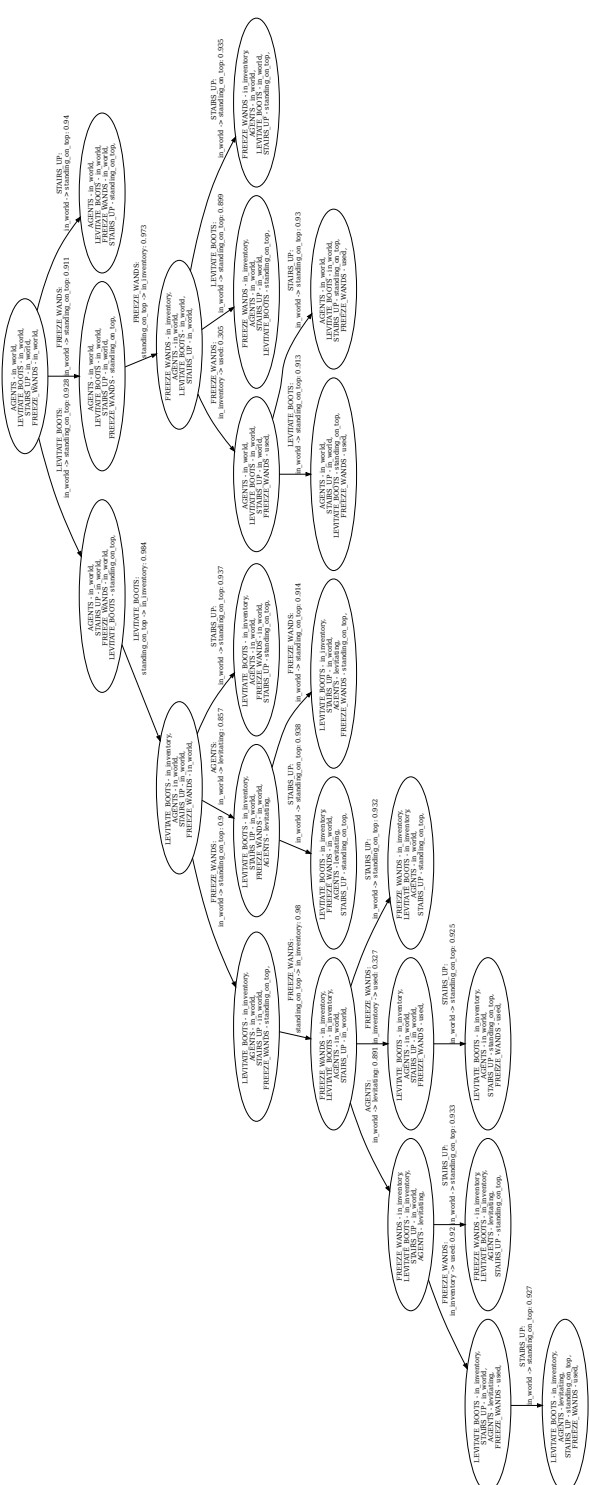

Figure 34: Extracted internal world model by starting traversal from a given initial state. Only the most relevant objects are included for clarity. The plan required to solve the Levitate-then-Freeze task involves going from the top (root) state to the bottom left state.

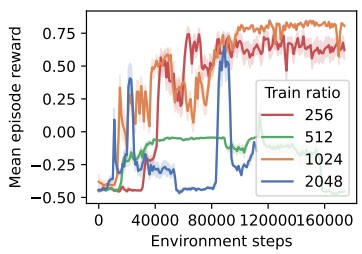 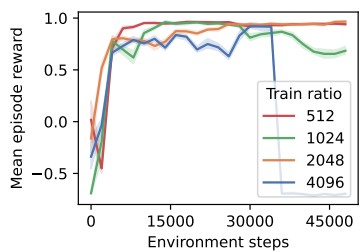

(a) Training from scratch (no pre-training)

(b) Fine-tuning after reward-based pretraining

Figure 35: Effect of Dreamer-v3 training ratio (amount of training to data collection) on performance, for training from scratch in `MiniHack-Freeze-Wand` (left), and for fine-tuning in `MiniHack-Levitate-Boots` from reward-based pretraining in `MiniHack-Freeze-Random`.

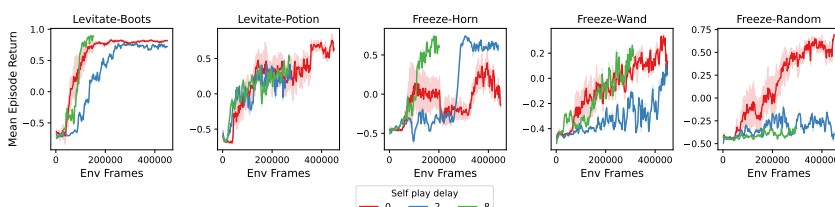

Figure 36: Muzero runs in the Minihack environments, with different self-play delay lengths

| Hyperparameter | | Value |
|---|---|---|
| Data Collection | Number of workers | 2 |
| | Number of env per worker | 10 |
| | Rollout length | 128 |
| | Recurrence length | 2 |
| Optimization | Optimizer | Adam |
| | Model lr | 1e-4 |
| | Reward scale | 1.0 |
| | Batch size | 4096 |
| | Shuffle minibatches | False |

Table 10: PPO Parameters

