# OpenReview forum: "Efficient Exploration and Discriminative World Model Learning with an Object-Centric Abstraction"
_ICLR.cc/2025/Conference — ICLR 2025 Poster_

### Official Review · Reviewer_5fJW · 2024-11-02

**Soundness:** 2
**Presentation:** 3
**Contribution:** 2
**Rating:** 6
**Confidence:** 4

**Summary:**

Summary: The authors propose a method to perform exploration and planning in an abstract MDP which they denote as Ab-MDP.
This assumes the  a set of abstract states is given, where the abstract state consists of an item id and a specific attribute for the object. The authors define as a competent behaviour, when the corresponding attribute of an item changes in an admissible manner within k steps. The authors train a forward model that predicts the probability i.e. the next
possible state from the competent behaviour. For finding the correct set of behaviours, MCTS together with a count based intrinsic reward is used. During inference time, the behaviours can then simply be found
by applying the Dijkstra algorithm on the probabilities of the competent behaviours put out by the forward model. The authors show that their model trained on the Ab-MDP outperforms methods trained on a low-level MDP as well
as known model-based methods such as Dreamer on the Ab-MDP.

Overall, I find the proposed abstraction an interesting and necessary view for exploration and policy learning in reinforcement learning. However, I feel that this work makes very strong assumptions on the needed abstractions for the proposed Ab-MDP, being tailored to the tested environments.
One thing that would mitigate this is a discussion on how the proposed item id, item attribute abstraction can be a general representation for other problems. For instance the attributes (in world, standing on, in inventory) may be meaningless in other environments or be a level of abstraction
That is too coarse to learn a good policy. How do we choose a good level of abstraction? Or what are desirable attributes that may allow for learning a good policy in most cases? I think this is important to make the Ab-MDP formalism more generalisable.

**Strengths:**

- Learning from more complex and abstract behaviours is an important problem in reinforcement learning, especially since the current focus so far as the authors state is mainly on problems that can be solved without much planning.
- The authors propose a straightforward architecture to explore and plan from discrete sets, i.e., abstract states showing they perform better than models that are not tailored to this abstract MDP.
- The paper is well written and the provided illustrations help in making the method understandable. I also found the ablations to be insightful justifying the components of the method.
- I think the results are promising, showing that the definition of a problem can be just as important as the underlying methods to solve it.

**Weaknesses:**

- The paper makes a lot of assumptions about the nature of how attributes and behaviours of the states are encoded. While a thorough explanation is provided for the given environments, it seems unfeasible to do this exhaustively for more complex environments.  I feel there needs to be a more general analysis of what constitutes an item attribute, I.e. a good abstraction level for an Ab-MDP.  In my view, when defining a new framework such as a variation of an MDP, it should define a general way to see the MDP that either is a useful abstraction or is tailored to a specific set of problem. While I see that the authors attempt this, it seems more that the authors restrict the MDP in a specific way to work with the given environments.

- At inference time, the authors rely on Dijkstra to plan towards a specific state with their trained forward model. While this seems feasible for static environments, it seems less feasible for procedural environments, where, e.g., the position of objects could change. At the moment it seems to me that with the current environments the forward model has to learn all possible combinations of item id and item attributes. Could you illustrate how performance degrades w.r.t to how much the forward model is trained, i.e., how much the encoder overfits to the environments?

- In the end, the authors perform a lot of manual feature engineering to make both environments fit in the Ab-MDP paradigm. In this way the contribution becomes really about learning a good model for this feature engineered solution. Although I do value the attempt of a world model paradigm that is based on discrete states (see positives).

**Questions:**

- How do you train Dreamer-v3 on the proposed environments? I would assume performance to be worse for the proposed Ab-MDP since for instance the latent space bottleneck in Dreamer is tailored to continuous and not discrete states.

---

> ### Author Response · Authors · 2024-11-24
>
> We thank the reviewer for the very thoughtful review. We are glad to find our work deals with an important problem in RL, with promising results, and is well written. We appreciate the reviewer’s astute summary that “the definition of a problem can be just as important as the method to solve it”.
>
> Below, we hope to address some of the weaknesses and questions for further discussion.
>
> ## Response to Weaknesses (W) and Questions (Q)
>
> > W1: The paper makes a lot of assumptions about the nature of how attributes and behaviours of the states are encoded. While a thorough explanation is provided for the given environments, it seems unfeasible to do this exhaustively for more complex environments. I feel there needs to be a more general analysis of what constitutes an item attribute, I.e. a good abstraction level for an Ab-MDP. In my view, when defining a new framework such as a variation of an MDP, it should define a general way to see the MDP that either is a useful abstraction or is tailored to a specific set of problem. While I see that the authors attempt this, it seems more that the authors restrict the MDP in a specific way to work with the given environments.
>
> The Ab-MDP’s choice of modeling sets of (item id, item attribute) gives a flexible abstraction framework, as items and attributes can be flexibly defined. In practice, items usually correspond to actual objects in the world. This is not an uncommon practice, for instance, various works from already work with item sets as a level of abstraction (e.g. [LOCA20, NCS23]).
>
> We also point the reviewer to Appendix section B.1.4 for some potential example application of Ab-MDP to various robotics environments.
>
> > W2a: At inference time, the authors rely on Dijkstra to plan towards a specific state with their trained forward model. While this seems feasible for static environments, it seems less feasible for procedural environments, where, e.g., the position of objects could change.
>
> Our environments _are_ procedure across episodes: for example, in the Minihack Freeze-Wand environment, one out of the 27 different types of wands are randomly spawned in each episode. The location of all objects and the agent are also random across episodes.
>
> In the scenario that item attributes change _within_ an episode (independent of the agent’s perturbation), we think re-planning (which MEAD does) would deal with this.
>
> > W2b: At the moment it seems to me that with the current environments the forward model has to learn all possible combinations of item id and item attributes.
>
> It is not necessary to learn all possible combinations for our model to work. Generally speaking, any model-based method needs to model the distribution $P(X_{t+1} | X_t, b_t)$ (i.e. put density over the entire state space $\mathcal{X}$). We make specific assumptions about this distribution (attribute change for a single item in $X_t$), which allow us to model this distribution using a discriminative model. We believe our learning efficiency (Figure 4), positive transfer (Figure 5a,b), and discriminative model ablation (Section 4.4.1) results demonstrate that compared to common baseline methods, we can in fact do better with having seen the same amount of (item id, item attribute) combinations.
>
>
> > W2c: Could you illustrate how performance degrades w.r.t to how much the forward model is trained, i.e., how much the encoder overfits to the environments?
>
> We perform an ablation by training the forward model with a different number of optimizer steps between data collection (note we reset the weights of our model and re-fit via sampling from the full dataset after each data collection; for ablation on this see Figure 28c). We see in Figure 30 that as long as we train for enough steps, the accuracy stays high (Figure 30 left), and the planning performance is good (Figure 30 right). If the model is insufficiently fit then accuracy decreases and planning gets worse.
>
> We further note that our model is not “overfitting” to the environment: this is evident by the positive transfer results in Figure 5a and 5b, where our model can zero- and few-shot transfer to unseen environments / objects.
>
> ---
>
> [LOCA20] Locatello, Francesco, et al. "Object-centric learning with slot attention." Advances in neural information processing systems 33 (2020): 11525-11538.
>
> [NCS23] Chang, Michael, et al. "Neural constraint satisfaction: Hierarchical abstraction for combinatorial generalization in object rearrangement." arXiv preprint arXiv:2303.11373 (2023).

---

> > ### Author Response · Authors · 2024-11-24
> >
> > ## Response to Weaknesses (W) and Questions (Q), Part 2
> >
> > > In the end, the authors perform a lot of manual feature engineering to make both environments fit in the Ab-MDP paradigm. In this way the contribution becomes really about learning a good model for this feature engineered solution. Although I do value the attempt of a world model paradigm that is based on discrete states (see positives).
> >
> > Please see Global response A.1 for discussion on the assumption of the object-centric map used to build the Ab-MDP. We appreciate the perspective of the discrete-state paradigm being useful and agree that “definition of a problem can be just as important as the underlying methods to solve it” -- and we’d argue that the object centric definition (at an abstracted level) is a useful definition applicable to many settings.
> >
> >
> > > Q1: How do you train Dreamer-v3 on the proposed environments? I would assume performance to be worse for the proposed Ab-MDP since for instance the latent space bottleneck in Dreamer is tailored to continuous and not discrete states.
> >
> > In short, Dreamer-v3 (along with all other methods trained in the Ab-MDP) take in sets of vectors as input. We run ablations for e.g. its model size and training training to data collection ratio and pick the strongest setting. We provide more details in Appendix G.2, with hyperparameters used, and also added an additional figure (Fig 35) ablating Dreamer's training ratio.
> >
> > In general we agree with the observation that Dreamer is well-tailored to a variety of continuous (in particular image-based) settings, and importantly, do not have useful inductive biases specific to solving the Ab-MDP (such as discriminative modeling). Our claim was not that MEAD is more general than Dreamer, but that MEAD solves the Ab-MDP well, and the Ab-MDP is a useful framework. There are methods that are more focused on “structured” MDPs similar to the Ab-MDP, and we compare MEAD against them in appendix section F.1.3 and show MEAD out-performing them. Finally, the reviewer may also find our newly added Table (5) interesting, which breaks down the main modeling choices between different model-based methods.

---

> ### Comment · Reviewer_5fJW · 2024-11-25
>
> I want to thank the authors for the detailed response!
>
> I feel my concerns have been addressed properly, however I still feel that the approach to define hard coded item attributes for a specific environment seems not very generalisable.
>
> I do appreciate the authors' example applications in Table 1, but then I wonder why not at least 1 of these environments was tested as well, which would strengthen the contribution. In this way, it would be clearer that the approach works solidly and working out the abstraction is a separate area of research that can be left for future work.
>
> I maintain my point that I find a world-model based method for discrete states very interesting and think this is the main contribution I value the most of this work. I therefore raise my score to marginally accept this paper.

---

> > ### Author Response · Authors · 2024-12-04
> >
> > We are pleased that the reviewer’s concerns have been addressed and appreciate their recognition of discrete state world models as an interesting direction!
> >
> > We agree that additional evaluation environments could further strengthen the work. Our decision was to focus on environments that emphasize the challenges at the abstract level (even with a given abstraction encoder). For instance, in the crafting environment, discovering the crafting tree has non-trivial item dependencies which some (abstract MDP) baselines fail to solve (Figure 23). Similarly, in MiniHack, the presence of many similar object types tests the model's ability to generalize effectively. Nevertheless, while we are confident in the ability of our current approach to work in even more environments, more (experiments) is almost always better. Additionally, we acknowledge that having a _fully automatic_ way of acquiring the abstract map for arbitrary environments remains an open question -- one that provides an exciting direction for future work.
> >
> > Overall, we are glad the reviewer finds the work compelling and are encouraged by their positive reception of the paper and decision to raise the score. We sincerely thank the reviewer for their time and insightful feedback, which has undoubtedly strengthened this work.

---

### Official Review · Reviewer_UYL3 · 2024-11-03

**Soundness:** 3
**Presentation:** 3
**Contribution:** 3
**Rating:** 6
**Confidence:** 3

**Summary:**

This work presents MEAD, a model-based planning method that learns a semantic reward-free world model in an abstracted MDP. The learned model is then combined with standard search algorithms to generate goal-reaching trajectories.  The abstracted MDP comprises of an object-centric mapping, where a state consists of a discrete set if items, each associated with a set of attributes. Notably, this work focuses on learning a discriminative world model, which predicts the probability of reaching a particular state, given an action (defined in the paper as behaviour) and current state. Results on 2-D crafting and Minihack show that MEAD significantly outperforms existing baselines.

**Strengths:**

1. **Clear writing and presentation**: The paper is well-written and generally easy to follow. It provides the right intuition and effectively builds up motivation where needed. The related work thoroughly covers existing work in the area.
2. **Strong and Extensive Results**: The proposed method performs strongly against competitive baselines and it can have significant advantages in settings where the assumptions of the work are satisfied. The paper also has a comprehensive set of experiments and analysis to support its claims. Notably, the paper has numerous ablations to understand the importance of the different components of the proposed method, which was enjoyable to read.

**Weaknesses:**

1. **Limited Applicability**: The presented method assumes access to object-centric maps, low-level behavior policies and discrete space for attributes. Many RL environments do not provide access to all of the above, and while it is possible to learn these policies/maps, it might require a lot of overhead.
2. **More clarity in results**: The paper does not do a good job of introducing the baselines and only describes them superficially in the main paper. More detailed descriptions will help increase clarity.

**Questions:**

1. The underlying assumptions of Section 4.1 need to be clarified further. If Dreamer and other baseline policies also assume access to these behaviors and operate in the same abstract MDP, then what are the most likely for the differences in sample efficiency. Inference for MEAD is done using Dijkstra's, how is it done for the other methods? Some analysis of the results in *Section 4.1* would be helpful .

2. In general, it would be useful to have 2 sentence descriptions of the baseline methods or a table that shows the high-level differences between MEAD and baselines (access to behavior policies, abstract world model, inference-time strategy, etc.,). This will help increase clarity as currently it is a bit difficult to keep track of the differences.

3. In the transfer experiments, Dreamer and other baseline struggles, and the authors hypothesize that this may be due to a distribution shift. Why does MEAD not suffer from a similar distribution shift? Does this have to do with the discriminative model it learns or because of other assumptions in the work.

4. Section 4.3.1 discusses learning all low-level behaviours. However, I imagine that in most environments, the number of behaviors is extremely large ( $N*m$) and most of the them are incompetent. How much overload does this actually add in practice? What are the total number of low-level policies learned and what fraction of the behaviors are competent (success probability above some threshold)? What are the cumulative training steps (and wall-clock hours) spent in learning these behaviors, and how do they compare to the metrics for the world-model training?

5. How do you expect this method to perform when attributes are continuous (eg., speed, position, etc.,)? Continuous attributes can be commonly found in many robotic domains, and this work relies on having a discrete set of states.

---

> ### Author Response · Authors · 2024-11-24
>
> We thank the reviewer for the insightful feedback. We are very encouraged to find that our paper is clearly presented with a thorough respect for related works, and demonstrates strong empirical results that are extensively ablated and analyzed.
>
> Below, we hope to address the questions.
>
> ## Response to Questions (Q)
>
> > Q1: What are the likely explanations for differences in sample efficiency between MEAD and baselines?
>
> We believe this is due to how MEAD models $P(X_{t+1} | X_t, b_t)$. Specifically, we make specific assumptions about this transition (attribute change for a single item in $X_t$, see Equations 2 and 3), which allow us to use a *discriminative* model. We argue that this is a _useful_ assumption as at an abstract level, many things in the world _do_ in fact change only locally at a time. We are also not the first to make such an assumption (for e.g. [NCS23]).
>
> On the other hand, Dreamer makes more general assumptions and model $P(X_{t+1} | X_t, b_t)$ *generatively* with little restrictions put on $X_{t+1}$. We directly show the difference in this modeling choice in Ablation Section 4.4.1, and show visualizations of the learned embedding space within the discriminative model in Appendix F.5.
>
> > Q2: Useful to have descriptions of the baseline methods. This will help increase clarity as currently it is difficult to keep track of differences.
>
> Many thanks for the recommendation. We have updated our draft to include both a short description of baselines at the start of Section 4 and an Appendix Table 5 to fully outline their main differences. We hope the reviewer finds this helpful for improving clarity.
>
> > Q3: Why does MEAD not suffer from distribution shift? Does this have to do with the discriminative model it learns or because of other assumptions.
>
> The reviewer is correct, we believe the greater robustness to distribution shift is due to discrimination vs. generation. Specifically, the forward model in MEAD only needs to identify whether a change in a single item is possible, which is easier to generalize given minor changes in other items. On the contrary, “generative” forward models (e.g. Dreamer) needs to model a distribution over the full set of items and is much harder. This is indirectly shown in Figure 7 ablation.
>
>
> > Q4: Section 4.3.1 discusses learning all low-level behaviours. However, I imagine that in most environments, the number of behaviors is extremely large (N∗m) and most of the them are incompetent. How much overload does this actually add in practice? What are the total number of low-level policies learned and what fraction of the behaviors are competent (success probability above some threshold)? What are the cumulative training steps (and wall-clock hours) spent in learning these behaviors, and how do they compare to the metrics for the world-model training?
>
> Our paper focuses mainly on what you do once you have a set of competent low-level policies. To get such policies, one could use behaviour cloning, hand-coded controllers, RL, etc.. We demonstrate one example here (of using RL) to learn them. Further, while $N \times M$ is indeed large, policies that change attributes are often generalizable. For instance in robotics, for the attribute change of “standing_over” to “in_inventory” (i.e. a “pickup” policy), typically people will learn one grasping policy and condition it on the object, rather than learning N seperate policies. [OKR24].
>
> In our experiments, the specific number of competent low level policies depends on the environment. For instance in the Freeze-Wand environment (27 different types of wand), over 370 unique goals were attempted over the course of low level training, while only around 60 unique goals were achieved at least once. Training required ~ 20 million frames and takes 3-4 hours with a performant goal-conditioned PPO implementation. This is more samples than what is needed to train MEAD, albeit this cost is incurred by all methods we compare that use low level neural policies. All in all, the purpose of this section was mainly to reassure that these policies do not need to be perfect or hard-coded for MEAD to work.
>
>
> > Q5: How do you expect this method to perform when attributes are continuous (eg., speed, position, etc.,)? Continuous attributes can be commonly found in many robotic domains, and this work relies on having a discrete set of states.
>
> Please see Global response A.2.
>
> ---
>
> [NCS23] Chang, Michael, et al. "Neural constraint satisfaction: Hierarchical abstraction for combinatorial generalization in object rearrangement." arXiv preprint arXiv:2303.11373 (2023).
>
> [OKR24] Liu, Peiqi, et al. "Ok-robot: What really matters in integrating open-knowledge models for robotics." arXiv preprint arXiv:2401.12202 (2024).

---

> > ### Comment · Reviewer_UYL3 · 2024-11-28
> >
> > I thank the authors for their response. While the results of this paper are noteworthy, I think the assumptions made in the proposed framework limit its applicability. The framework requires access to competent low-level policies which might be hard or expensive to obtain. In environments with a large number of objects and attributes, the number of low-level policies needed might be extremely larger. Similarly, it is unclear whether this method can be adapted to continuous spaces (although the authors do present some possible ways to do this). Finally, some of the improvements of this framework seem to be a result of using a discriminative formulation instead of a generative one. The benefits of this are definitely interesting for the community, but there exists a good amount of prior work (in unrelated domains) on the differences between generative vs discriminative learning.
> >
> > As a result, I will maintain my current score.

---

> > > ### Author Response · Authors · 2024-12-04
> > >
> > > We thank the reviewer for their response, and the recognition that our results are noteworthy.
> > >
> > > Regarding the **reliance on competent low-level policies**, we agree that _if no other ways of acquiring policies are available_, training low level policies using RL can be expensive. However, in this scenario, **the alternative options are similarly bad**. For instance, training monolithic task-specific policies with RL will take just as long if not longer as separately training each compostable sub-policy. The monolithic policy is also harder to train as the reward signal is more sparse, nor is it re-usable to solve other tasks in the same environment. Notably, **MEAD does not require _all_ low level policies to be pre-trained to work**. While this may limit the space of what is discoverable by the world model, it still enables "partial" exploration to abstract states reachable by the currently competent behaviours. We will provide a measured discussion of the above and acknowledge these limitations in the discussion section.
> > >
> > > Additionally, while we do not claim to contribute in a general way to discriminative vs. generative learning, we believe **applying discriminative modeling to a setting commonly tackled by generative methods (world modeling) offers a novel perspective**. With an abstraction which enables discriminative modelling, we see various learning efficiency and transferability benefits. As reviewer 5fJW notes: _the definition of a problem can be just as important as the underlying methods to solve it_. This is the purpose of abstractions, and what we hoped to demonstrate in our work.
> > >
> > > We once again thank the reviewer for their valuable feedback, which has undoubtedly strengthened the paper.

---

### Official Review · Reviewer_ccnG · 2024-11-04

**Soundness:** 3
**Presentation:** 3
**Contribution:** 2
**Rating:** 6
**Confidence:** 4

**Summary:**

This paper proposes a method for efficient exploration and world model learning in reinforcement learning (RL) by leveraging an object-centric abstraction. The authors define an Abstracted Item-Attribute Markov Decision Process (Ab-MDP), where abstract states are sets of items and their attributes, and actions are abstract behaviors corresponding to object-perturbing policies. They introduce MEAD (Model-based Exploration of abstracted Attribute Dynamics), a fully model-based algorithm that learns a discriminative world model predicting the success probability of abstract behaviors. MEAD uses count-based intrinsic rewards and Monte-Carlo Tree Search (MCTS) for efficient exploration and plans using Dijkstra's algorithm to reach goal states. The method is evaluated on a suite of 2D crafting and MiniHack environments, demonstrating significant improvements in sample efficiency and performance over state-of-the-art baselines. The authors also show that MEAD transfers well to new environments and tasks and discuss how to learn the object-centric mapping and object-perturbing policies when they are not provided.

**Strengths:**

1. Novelty: The approach proposed improves exploration and world model learning in RL by utilizing object-centric abstractions and discriminative modeling.
2. Efficient Exploration: MEAD adopts a discriminative world model and countbased intrinsic reward for efficient exploration.
3. Experimental Results: Plots and explanations are detailed and clear.
4. Transfer and Generalization: The paper shows MEAD can transfer zero-shot and few-shot to new environments and object types indicating of its generalization ability.
5. Interpretable World Model: Object-centric abstraction naturally can lead to an interpretable world model which can be helpful for understanding agent behavior.

**Weaknesses:**

1. Assumption of Given Abstraction: The approach inherently assumes access to an object-centric mapping and competent object-perturbing policies. This limits the applicability in scenarios where getting these mappings can be cumbersome.

2. Abstraction Learning: The authors discuss learning the object-centric mapping and the policies. However, more details and empirical validation would strengthen this section.

3. Experimens: It is not clear if the experiments using PPO and Dreamer-v3 are applied to Ab-MDP or to the MDP setting. It is mentioned that these methods are used in Ab-MDP, but there also is "we also show the final performance of state-of-the-art exploration  methods in the low level MDP" in the caption for Figure 4.
Additionally, there is no comparison with any of the methods that are closer to the proposed method (a good number of them are mentioned in the Related Work section). Also, for the comparison with PPO and Dreamer-v3, the proposed method is not evaluated in the settings and environments in which the baseline methods are mostly evaluated for their performance. Therefore, it is unclear how the proposed method is generalizable over more complex environments and the comparison do not sound quite fair.

4. Clarity: Certain parts of the paper and the overall flow of information can be improved and clarified. For example, the paragraph starting at line 161 and ending in line 167 could be revised to convey the message more clearly. Additionally, the explanation of T(X′|X, b) in lines 109-111 could be rephrased for better clarity. Here are some errors I noticed: in line 316, "the agent observe" should be corrected to "the agent observes," and in line 331, "themin" should be separated as "them in."

**Questions:**

1. Could you provide more details on how the object-centric mapping \( M \) is learned in practice, especially in complex environments and the environments where such mappings are not readily available? How does the performance of MEAD depend on the quality of this mapping?

2. What are the computational requirements of MEAD compared to the baselines? For example, how does the planning  (MCTS and Dijkstra's algorithm used) affect it and is it feasible in more complex environments with larger state space and action space?

3. How does your method compare to the methods closer to yours, such as the ones mentioned in your Related Work section?

---

> ### Author Response · Authors · 2024-11-24
>
> We thank the reviewer for the insightful feedback. We are glad to see the reviewer think our method is novel and clearly presented, and that MEAD demonstrates strong results in exploration, transfer and generalization, and has the added benefit of being an interpretable world model.
>
> Below, we hope to address some of the weaknesses and questions for further discussion. We've added additional experiments and will refer to their exact locations in the updated pdf.
>
> ## Response to Weakness (W) and Questions (Q)
>
> > W1: The approach inherently assumes access to an object-centric mapping and competent object-perturbing policies.
>
> Regarding getting mapping: we acknowledge this limitation. Please also see Global Response A.1 for a further discussion on the object-centric mapping.
>
> Regarding competent policies, we show in section 4.3.1 that we can learn them using reinforcement learning as long as we have access to the object-centric mapping (albeit less sample-efficiently). In practice it is also often easier to get sub-policies, such as by imitation learning [GUP19], than composing them together.
>
> > W2a and Q1: Details of training the object-centric mapping. How does MEAD depend on the quality of this mapping.
>
> Per section 4.3.1, we learn the object centric mapping via supervised learning, assuming a labeled dataset of $(S, X)$ pairs can be collected. Given this dataset, we train an encoder which takes the low level (image-like) observation as input, and outputs a set of vectors. We can train this with supervised learning to predict the set of (item id, item attribute) categories given a low level observation. We have added more detailed training and architectural details for this in Appendix section F.2.2.
>
> Additionally, we conducted an experiment by varying the training dataset size to investigate the effect of an *inaccurate* encoder map (M) on the downstream MEAD performance. We observe with less training data, the (held out) accuracy of the map decreases as expected (Figure 25, left). Nevertheless, we find that even with imperfect map, MEAD is surprisingly robust and able to solve the task even with ~60% prediction accuracy (Figure 25, right).
>
> The point of the experiment is to demonstrate MEAD can work with a sufficiently well-learned mapping. As stated clearly, this work is not focused on unsupervised learning of such a mapping, although how to *use* such a mapping is not clear and the point of this current work.
>
> > W2b: More details on policies
>
> Low level policies can be trained as goal-conditioned policies (conditioned on a specific behaviour), as described in section 4.3.1. We see from Appendix Section D.2.2 that for “imperfect” low level policies (i.e. do not 100% lead to the desired transition), MEAD is still robust. In fact, MEAD directly models the probability of policy success, and Dijkstra plans a path to maximize this probability.
>
> > W3: Not clear if PPO and Dreamer-v3 are applied to Ab-MDP or to the MDP setting… The proposed method is not evaluated in the settings and environments in which the baseline methods are mostly evaluated for their performance. Therefore, it is unclear how the proposed method is generalizable over more complex environments and the comparison do not sound quite fair.
>
> All methods (PPO and Dreamerv3) are applied to the Ab-MDP (as we state in L124), with the only exception being the black triangles / stars in Figure 4. We apologize if this point was not clear and have updated Figure 4b's legend. Specifically, the point of Figure 4b is three-fold:
>
> - We wished to show that “vanilla” standard methods (e.g. PPO) can be applied to solve an Ab-MDP
> - When solving the task in the Ab-MDP (as opposed to the low level MDP), regardless of method, we get a large efficiency gain (e.g. comparison between vanilla PPO in Ab-MDP in yellow, and state-of-the-art exploration methods + PPO in the low level MDP, as black stars and triangles)
> - Finally, MEAD exploits the structure of the Ab-MDP better and is the most efficient method within the Ab-MDP
>
> The point here is that Ab-MDP is a useful framework, and MEAD an efficient method to solve it. We do not claim MEAD as a general MDP solver (as is the case for e.g. PPO), and therefore do not compare MEAD with other methods in low level MDPs that PPO and Dreamer usually operate in.
>
> > Clarity
>
> We are grateful for the clarity comments. We have updated the new pdf to reflect this (L163, L111, L320). We hope the reviewer find the changes suitable and we are open to additional comments.
>
>
> ---
>
> [GUP19] Gupta, Abhishek, et al. "Relay policy learning: Solving long-horizon tasks via imitation and reinforcement learning." arXiv preprint arXiv:1910.11956 (2019).

---

> > ### Author Response · Authors · 2024-11-24
> >
> > ## Response to Weakness (W) and Questions (Q), part 2
> >
> > > Q2: What are the computational requirements of MEAD compared to the baselines? For example, how does the planning (MCTS and Dijkstra's algorithm used) affect it and is it feasible in more complex environments with larger state space and action space?
> >
> > MCTS is an “anytime” algorithm, thus its compute budget is as much as we allocate it. In practice we run 16 simulations up to a depth of 4 (we ablated this choice in Figure 8).  We run Dijkstra during “exploitation” phase as the resulting abstract world model is fairly sparse and we do not need to search every branch. If this is not the case we can also run MCTS which is well suited to complex state spaces.
> >
> > In practice, a run with MEAD on CPU and Dreamer-v3 on GPU takes approximately the same time (experiments takes ~24 hours to run 100-200k training env steps, including intervalled evaluation), although these details can differ based on implementation details, optimizations and hyperparameters.
> >
> > We should note that our environments are not necessarily non-complex. For instance, in the compositional planning environment (Fig 5c), all baseline methods struggle other than MEAD. Figure 34 gives an idea of an (subset of an) abstract state graph that is fairly complex which MEAD learns in an unsupervised way. However, for Dijkstra to work efficiently, we would likely want an internal graph that is not too densely connected. We argue a densely connected graph is *not* the case for the real world at the object level, but nevertheless, if Dijkstra becomes too inefficient, we can use MCTS during evaluation time as well.
> >
> >
> > > W3b and Q3: There is no comparison with any of the methods that are closer to the proposed method (a good number of them are mentioned in the Related Work section).  How does your method compare to the methods closer to yours
> >
> > We already compare against the attribute planner [ZHANG18] which is a non-parametric version of our model (already referenced in related works and in section 4.4.3, result in Figure 17).
> >
> > We ran additional experiments against the NCS model as described by [NCS23], which also uses a similar “item id, item attribute” set MDP structure. We observe in Figure 23b that NCS completely fails as it only models attribute change without considering item identity and plans in a greedy fashion.
> >
> > We run additional ablation for a model which uses Dijkstra to plan at eval time (as we do), but uses K-means to learn an internal graph for planning between centroids (as NCS does). We observe that in MiniHack, this works but is slightly worse than MEAD (Figure 23b), and in 2D crafting this completely fails (Figure 23a). The reason for the latter failure is due to the 2D-crafting crafting tree violating NCS’ assumption of the attribute change being independent of any other items’ attributes (discussed in Appendix B.1.3). These experiments are described further in Appendix F.1.3.
> >
> > We have also updated Appendix B and added Table 5 which summarize the main difference between model-planning methods. We also updated the related works section to additionally mention these comparisons (L466, L470) for greater clarity.
> >
> > ---
> >
> > [ZHANG18] Zhang, Amy, et al. "Composable planning with attributes." International Conference on Machine Learning. PMLR, 2018.
> >
> > [NCS23] Chang, Michael, et al. "Neural constraint satisfaction: Hierarchical abstraction for combinatorial generalization in object rearrangement." arXiv preprint arXiv:2303.11373 (2023).

---

> > > ### Comment · Reviewer_ccnG · 2024-11-25
> > >
> > > Thank you for your clarification and addressing the questions and concerns mentioned in the review.
> > >
> > > By more complex environments I meant the environments that involve a large number of objects and object attributes. Also, by more similar methods, I meant comparing with work using different high level representations. However, I mostly got the answers to my questions and concerns.
> > >
> > > I will update the overal score according to the new revision. Good luck!

---

> > > > ### Author Response · Authors · 2024-12-04
> > > >
> > > > We sincerely thank the reviewer for their feedback and are pleased to see their score improve following our revisions and clarifications.
> > > >
> > > > Regarding the **complexity of environments**, we agree increasing the number of items and attributes would stress-test model-based methods (ours and others) in an interesting way. We do believe MEAD can scale _with_ an efficient search method (e.g. MCTS). As it is currently, we have a large number of _possible_ abstract states (e.g. in MiniHack tasks have ~ $N=13$ items present, $M=6$ possible attributes, thus $M^N \approx 13$ billion possible abstract states). MEAD reduces search complexity by (i) only considering single item-attribute changes (at most $N \cdot M$ branches to expand), and (ii) efficiently learning a good world model, which sparsifies the search tree. Nevertheless, we acknowledge the scaling of search efficiency to even larger state spaces is a currently untested question, and we will add a measured discussion about this in the discussion section.
> > > >
> > > > We also agree that **comparing different high-level representations** is an interesting direction for future work, requiring non-trivial experimental designs to carefully tease out the effect of the abstraction quality, the world model, and planning algorithm. To the extent that other methods are compatible with an item-attribute set representation, we have tried our best to evaluate these methods (namely, attribute planner and NCS), to specifically tease out the effect of model and planning, while keeping the abstraction quality similar.
> > > >
> > > > Overall, we are pleased that the reviewer’s concerns have been addressed and appreciate the constructive feedback, which has undoubtedly strengthened this work. Thank you again for your time and valuable insights.

---

### Official Review · Reviewer_74S3 · 2024-11-04

**Soundness:** 3
**Presentation:** 3
**Contribution:** 3
**Rating:** 8
**Confidence:** 3

**Summary:**

The proposed using an abstract model of the world to plan with, and presents a model learning algorithm along with it. The main selling point of these abstractions is that they are object-centric. They respect specific useful tools in the environments, and curate the agent’s learning to be around them. These objects and their attributes are also human-interpretable.
They show how this can lead to better performance than existing model-based methods, and perform an empirical study into the components responsible for this, along with listing the settings where it does work.

**Strengths:**

* Answers the questions they presented in the abstract and introduction.
* The Methods section is succinct and points the reader towards the relevant experiments, based on the topic / interest.
* Contributions are all listed usefully (and addressed)

**Weaknesses:**

* Abstractions are not discovered. It would be nice if this was told earlier in the paper.
* The Abstract MDP proposed here and the options framework are two different mathematical objects. The Ab-MDP cannot be interpreted as options. Under the options framework, the Ab-MDP _may_ be interpreted as a semi-MDP. As mentioned in the appendix, it is the behavior, b, can be interpreted as an option.
* In Section 4:
     - Paper doesn't explicitly mention how many seeds were used in the experiment. The appendix hints at least 3, but it would be good to know how many were used for each plot. This can contribute to the context with which the reader interpret the results, whilst also providing the paper a context with which to substantiate its claims.
     - Figure 4b: Which triangle is IMPALA, RND and ICM in each plot?
     - Figure 5: Why do none of the black lines have error bars? Are they single seed? If so, that would be useful to specify in the caption.
     - Figure 5a: Why does the blue line start high? Is this just a plotting artefact?
* [Very Minor] Last sentence of abstract: change wording to talk about “reinforcement learning” and “supervised learning”
* [Very Minor] Typo: line 325: “representatin”

**Questions:**

The relationship between the competence of a behaviour and the transition matrix was not made clear. Is the paragraph trying to say: a competent behaviour will have most/all of the probability mass concentrated on some other state s.t. X_{t+1} != X_t ?

---

> ### Author Response · Authors · 2024-11-24
>
> We thank the reviewer for the insightful review. We are glad to see that we’ve done a good job in addressing all claims and contributions we set out to answer, and that it is clearly presented.
>
> Below, we hope to address some of the weaknesses and questions for further discussion.
>
>
> ## Response to Weakness (W) and Questions (Q)
>
> > W1: Abstractions are not discovered. It would be nice if this was told earlier in the paper.
>
> We have revised our introductory section to clarify this, please see L064.
>
>
> > W2: Ab-MDP cannot be interpreted as options. Ab-MDP may be interpreted as a semi-MDP.. It is the behavior b that can be interpreted as an option.
>
> We agree the Ab-MDP may be interpreted as a semi-MDP, and behaviors can be interpreted as options. We believe our analysis in Appendix B.1.1 reflected this exactly. We updated our wording (L114, L434) to reflect our intention of interpreting the Ab-MDP under the Options *Framework*, which include both semi-MDPs and temporally extended actions.
>
>
> > W3: Seeds, legends and figures
>
> We apologize for the confusion. We have clarified all points in our updated draft:
> - We added a table with seeds in Appendix Section F.1, Table 6
> - The triangle actually refers to all three methods: which all get a reward of 0 and are overlapped. We have updated Figure 4b to make this less confusing
> - The black lines in Figure 5 are the same curves as Figure 4b (plotting just the mean for clarity). We have updated Figure 5 caption to clarify this.
> - Figure 5a: the blue line starts high as MEAD shows good zero-shot transfer from the sandbox environment to the new environment. This is likely because the sandbox environment spawns all objects with some probability, thus the agent has already seen the object (albeit in a setting where many other objects are present), and MEAD can effectively use this knowledge to transfer to environments that require using individual objects. The reviewer may also find our PCA analysis of MEAD’s representation space in Appendix F.5 and Figure 32b interesting, where we observe similar object-attribute pairs clustered together after learning.
>
>
> We have also fixed the typos, we are very grateful for the reviewer’s effort in pointing them out.
>
> > Q1: The relationship between the competence of a behaviour and the transition matrix was not made clear. Is the paragraph trying to say: a competent behaviour will have most/all of the probability mass concentrated on some other state s.t. X_{t+1} != X_t ?
>
> This is correct. More precisely, since behaviours are *defined* as *changes* to an item’s attribute $b = ( \alpha^{(i)}, \xi’ ) $, executing the behaviour policy may either lead to the change specified by the behaviour (i.e. the i-th item attribute changing to $\xi’$) from the current state $X_t$ to $X_{t+1}$, or not (no change or an un-specified change). Executing a competent behavior policy results in $P(X’ | X, b)$ having high probability where $X’$ contains the specified change (Equation 1).

---

### Official Review · Reviewer_eBxo · 2024-11-05

**Soundness:** 4
**Presentation:** 4
**Contribution:** 3
**Rating:** 8
**Confidence:** 4

**Summary:**

The paper proposes a method for solving particular object-centric abstract MDPs with discrimitive world models, count-based exploration and planning / shortest path methods.
The paper shows very strong empirical results on two problem domains, Minihack and Crafting. The paper also presents results on learning low-level behaviors and object-centric representation (in a supervised way).
Overall the paper contains intersting insights into model-based RL when applied to higher level abstraction.

**Strengths:**

- Overall, clearly presented
- very efficient method for solving high-level problems of a particular structure (Ab-MDP)
- study of different transfer aspects
- extensive comparison to strong method and impressive results
- comprehensive analysis with lots of additional details in the appendix

**Weaknesses:**

- The method needs quite strong assumptions on the high-level MDP, which are that usually only one element is changing or nothing changes. The authors also argue that replanning fixes some amount of violation of that assumption, but an environment where agent/object positions need to be modeled on the high level would not work from my understanding.

**Questions:**

1. I did not understand the exploration used in section 4.1:
You introduced the count-based exploration before, but that does not seem to be used here? The task return is when running Dijkstra, right? Is there an exploration phase here?
1. Can the authors comment on why they did not employ value iteration on the high-level instead of Dijkstra. That could deal with stochastic outcomes.
1. For the Dreamer and MuZero baseline: did you run enough / more updates per collected data in the transfer setting? If you reset the policy it needs to be learned again. Dreamer needs to run many updates using the dream-phase to rebuilt it. In this sense the comprison is a but unfair, as your are using run-time planning. I suggest to run Dreamer with a larger number of dream-updates with the new reward function before interacting with the environment.

## Comments:
- Very little detail about the training/architecture of the world model in the main paper, I suggest to at least mention the architecture and the loss in the main paper.
- The paper needs some careful proof reading:
    - Abstract: the last sentence : to reinforce learn and to supervise learn is an uncommon use of the words
    - first sentence of intro: have been major roadblocks: I think this is not correct. They have been topics of active research and many very effective solutions have been proposed. So I suggest to change that sentence
    - l086: eaching havin form
    - l143: more fully
    - l196: trainbale
    - l251: plot 95 confidence interval
    - l270: paragraph is missing details on how exploration is done etc.
    - l313/314, 331/332, 404/405

- In the related work, you might want to mention this paper:
Curious Exploration via Structured World Models Yields Zero-Shot Object Manipulation
https://openreview.net/forum?id=NnuYZ1el24C
that also shows that structured representations, world-model structure, and planning for exploration yield strong models and zero-shot capabilities (shown in manipulation).

---

> ### Author Response · Authors · 2024-11-24
>
> We thank the reviewer for the insightful and positive review. We are glad to see the paper contains interesting insights for abstract-level models, is empirically very strong, with extensive comparisons and analysis.
>
> Below, we address some of the weaknesses and questions for further discussion. We've added additional experiments and will refer to their exact locations in the updated pdf.
>
> ## Response to Weaknesses (W) and Questions (Q)
>
> > W1: Environment where agent/object positions need to be modeled on the high level would not work
>
> Please see global response A.2.
>
>
> > Q1: Exploration used in section 4.1.. Is there count-based reward used when running Dijkstra?
>
> There is not. Dijkstra is used to search over a sparser graph with a sparse reward function. We do not explore in this phase, but purely exploit to plan toward a goal state.
>
> > Q2: Why not employ value iteration on the high-level instead of Dijkstra? This could deal with stochastic outcomes.
>
> Value iteration (at least the standard version we are aware of) is too compute intensive. Consider an Ab-MDP with N=13 items and M=6 possible item attributes (as is the case in many MiniHack tasks), there are $M^N \approx 13 \text{ billion}$ possible abstract states $X$. Even though most state transitions are impossible, (naive) value iteration requires either iterating over or matrix-multiplying with all states, which is prohibitively expensive.
>
> Dijkstra avoids this by only locally searching *from* the initial states based on possible state transitions until a goal state is found (we do not search a transition if its success probability is low), and is efficient when the transitions are sufficiently sparse. In the case transition is not sparse, MCTS can be used (as is done for the exploration phase).
>
>
> > Q3: For the Dreamer and MuZero baseline: did you run enough / more updates per collected data in the transfer setting? If you reset the policy it needs to be learned again. Dreamer needs to run many updates using the dream-phase to rebuilt it. In this sense the comparison is a but unfair, as your are using run-time planning. I suggest to run Dreamer with a larger number of dream-updates with the new reward function before interacting with the environment.
>
> We ran an experiment examining the effect of different train ratio and decided on 1024 being the best choice for training from scratch (this also happened to be the highest ratio used in the original paper), see Figure 35a. We ran additional ablation for the transfer experiment and found that the train ratio had little effect on the initial learning speed, albeit a too-high ratio may destabilize training (Figure 35b). Details for this are in Appendix G.2.
>
> We also have the MuZero training ratio ablation in Figure 36, described further in Appendix G.3.
>
> All in all, we did not find that pushing the train ratio higher consistently helps with greater sample efficiency. Also note none of the settings are more sample-efficient than MEAD.
>
>
> ## Response to Comments
>
> We are deeply thankful for the writing suggestions! We have incorporated all changes in the attached updated pdf, as well as adding the new related work.

---

> > ### Comment · Reviewer_eBxo · 2024-11-25
> > **More Clarification**
> >
> > Thank you for your answers.
> > I am still puzzled by the lack of exploration in "Learning from scratch in single environments".
> > > There is not. Dijkstra is used to search over a sparser graph with a sparse reward function. We do not explore in this phase, but purely exploit to plan toward a goal state.
> > If you have not visited the abstract state yet, Dijkstra will not find a path, as there is non yet. How do you get to the exploratory behavior? I want to understand that properly and I also feel that the paper needs to be more clear about this point.
> >
> > In the general answer you write about continuous variables. However, I was also hinting at another limitation: if you have variables that are changing with high frequency, e.g. something like a position (can be discrete) the efficiency of the approach suffers as the ab-MDP becomes big.
> > I would like limitations on the abstraction are discussed in the limitations.

---

> ### Author Response · Authors · 2024-11-30
>
> ### Dijkstra & Exploration
>
> Many thanks to the reviewer for clarifying the question. We run our method MEAD in two _separate_ experimental "stages":
> - **Pure exploration phase**: we use the count-based intrinsic reward, and we use MCTS with the (partially learned) model to do search to find states with high intrinsic reward (Illustrated in Figure 3a). This encourages the agent to visit the state space widely and generate data. We use this data to train the model, and over time the model becomes more and more accurate, which in turn further helps with exploration. There is no notion of extrinsic reward or goal states here.
>
> - **Evaluation / exploitation phase**: given a model (trained up to some data budget from the pure exploration phase), we evaluate it by using Dijkstra to try to find a goal state (Figure 3b). It is entirely possible that if the goal state has not been visited, the transitions to the goal state will be deemed impossible by the model, and Dijkstra will not find the correct path to the goal state. This is what happens when the exploration phase has not been run for long enough (e.g. Figure 4b, with low env frames the episode reward is low). Also note we do not add the data generated in the eval phase into the buffer used for training the model.
>
> ---
>
> ### High Frequency Changing Attributes
>
> > if you have variables that are changing with high frequency, e.g. something like a position (can be discrete) the efficiency of the approach suffers as the ab-MDP becomes big
>
> Indeed, this is likely to be true and we will add this as a limitation. In general whether or not attributes like this is present would depend on the abstraction map $M$ itself, and the hope with (and general point of) abstractions is that things should happen at a slow(er) temporal resolution at a higher level. Nevertheless, we agree the Ab-MDP is likely less suitable for directly modelling these lower level attributes -- it's inductive biases make it less general for _all_ problems, but highly efficient for a (in our opinion still flexible and useful) subset of problems.
>
> ---
>
> We hope this addresses the reviewer's questions. Please let us know if you have more questions, and we apologize for this delayed response.

---

### Author Response · Authors · 2024-11-24
**Global Response (1/2)**

We thank each reviewer for their insightful engagement with our work. We are glad to see the reviewers think our method is novel and interesting (eBxo, ccnG), deals with an important problem (5fJW), has strong empirical results and extensive ablations (eBxo, ccnG, UYL3 5fJW), and is clearly presented with all claims and contributions substantiated (74S3, UYL3, 5fJW).

**We have addressed all concerns raised**. We are grateful to all reviewers for making our work stronger. **We have uploaded a new pdf containing new experimental results**, and we will refer to specific results *by their exact location in the new pdf*.

The major additions in the new pdf are:
- We show our method MEAD is **robust to inaccuracies in the learned object-centric map $M$**, and can solve the task even when the encoder accuracy is only ~60-70% (Appendix F.2.2, Figure 25)
- We compare against additional baselines that are similarly built for structured MDPs, and demonstrate that **MEAD consistently outperforms additional baselines** (Appendix F.1.3, Figure 23)
-  We provide additional ablations for the baseline methods, showing that higher training ratios do not aid Dreamer-v3 to learn or transfer better than MEAD (Figure 35)
- We outline main differences between world model learning approaches to **better dissect and give insights into modeling assumptions and inductive biases of existing works** (Table 5)
- Various clarification points and writing improvements

---

Below we address a few common concerns, before responding to each reviewers’ comments individually.


## A.1 Our assumption about the existence of an object-centric mapping

Reviewers mentioned the limitation of abstraction map $M: \\mathcal{S} \\rightarrow \\mathcal{X}$ being given, as well the specific form of the abstracted state as a set of (item id, item attribute), $X = \\{ (\\alpha^{(1)}, \\xi^{(1)}) , (\\alpha^{(2)}, \\xi^{(2)}), …\\}$. We fully acknowledged this limitation in the paper.

However, we point out that **having an object-centric encoder is a common assumption** in many works today. For instance, getting object-centric embedding from pre-trained foundational models (such as image and video segmentation models [XMEM22, SAM23]) are standard practices in *many* contemporary robotics works [ZHU23, QIA24, SHI24, XIE24]. The specific form of the object-centric abstraction as a *set* of items is also common [LOCA20], including sets that explicitly model id and attribute separately ([NCS23], discussion in Appendix B.1.3). Generally speaking, object centric representation (OCR) is an active sub-area of study and is fully complementary to MEAD: we show **MEAD is robust to imperfections in the object encoding** (Figure 25, Appendix F.2.2) and such **object representations will only improve with time**. Further, our Ab-MDP framework is more general than the environments presented in the paper: it can be defined with arbitrary item identities and attributes, we discuss some additional example applications in Appendix B.1.4.

While an object-centric assumption is common, our work investigates how to better make use of them: learn models discriminatively, use partially learned models for efficient exploration, and various empirical investigations such as transfer . Thus, we believe our work is **well-situated within and contributes to contemporary literature**.


## A.2 On our choice of discrete attributes and how to handle continuous attributes

The current Ab-MDP formulation does not consider continuous attributes. This is a trade-off: continuous attributes may be more general, but are difficult to search over. We argue that discrete attributes already cover a wide variety of tasks, and importantly, only the *abstract* attributes need to be discrete. The low level behaviour can deal with continuous attributes and actions (for examples of potential Ab-MDP applications to robotics tasks with continuous dynamics see Appendix B.1.4).

If somehow we absolutely need to model continuous attributes, we can nevertheless always discretize them [TRT21], and/or use various powerful discrete latent models [VQA17, VQB24]. This can be interesting future work.

---

> ### Author Response · Authors · 2024-11-24
> **Global Response (2/2)**
>
> [XMEM22] Cheng, Ho Kei, and Alexander G. Schwing. "Xmem: Long-term video object segmentation with an atkinson-shiffrin memory model." European Conference on Computer Vision. Cham: Springer Nature Switzerland, 2022.
>
> [SAM23] Kirillov, Alexander, et al. "Segment anything." Proceedings of the IEEE/CVF International Conference on Computer Vision. 2023.
>
> [ZHU23] Zhu, Yifeng, et al. "Learning generalizable manipulation policies with object-centric 3d representations." arXiv preprint arXiv:2310.14386 (2023).
>
> [QIA24] Qian, Jianing, et al. "Task-Oriented Hierarchical Object Decomposition for Visuomotor Control." arXiv preprint arXiv:2411.01284 (2024).
>
> [SHI24] Shi, Junyao, et al. "Plug-and-play object-centric representations from “what” and “where” foundation models." ICRA. 2024.
>
> [XIE24] Xie, William, et al. "DeliGrasp: Inferring Object Properties with LLMs for Adaptive Grasp Policies." 8th Annual Conference on Robot Learning. 2024.
>
> [LOCA20] Locatello, Francesco, et al. "Object-centric learning with slot attention." Advances in neural information processing systems 33 (2020): 11525-11538.
>
> [NCS23] Chang, Michael, et al. "Neural constraint satisfaction: Hierarchical abstraction for combinatorial generalization in object rearrangement." arXiv preprint arXiv:2303.11373 (2023).
>
>
> [TRT21] Janner, Michael, Qiyang Li, and Sergey Levine. "Offline reinforcement learning as one big sequence modeling problem." Advances in neural information processing systems 34 (2021): 1273-1286.
>
> [VQA17] Van Den Oord, Aaron, and Oriol Vinyals. "Neural discrete representation learning." Advances in neural information processing systems 30 (2017).
>
> [VQB24] Lee, Seungjae, et al. "Behavior generation with latent actions." arXiv preprint arXiv:2403.03181 (2024).

---

> > ### Author Response · Authors · 2024-12-04
> > **Final Global Comment**
> >
> > We are pleased that the reviewers continue to find our method interesting (5fJW) and our results noteworthy (UYL3). We believe **we have thoroughly addressed all concerns and questions raised by the reviewers**. _Additional baselines, ablations, and background information are detailed in the Global Response above._
> >
> > ---
> >
> > We believe incorporating key discussions from the review period into the main text will further enhance the paper and provide readers with a more nuanced understanding of our contributions and the perspective behind our work. Specifically, we plan to include discussions on the following topics:
> > - High-frequency attributes as a potential challenge for MEAD (eBxo).
> > - MEAD’s current search complexity and scalability to larger item-attribute sets (ccnG).
> > - Scenarios where no prior sub-policies (i.e. behaviours) are available for constructing the Ab-MDP, the cost of training all behaviours using RL, and MEAD’s handling of an incomplete behaviour set (UYL3).
> > - The perspective that applying discriminative modelling to world modelling offers a novel and interesting perspective and direction for additional work (UYL3).
> >
> > ---
> > As the discussion period concludes, we sincerely thank all five reviewers for their valuable feedback. The revisions made during this period, along with the planned updates, will undoubtedly strengthen the paper.

---

### Meta-Review · Area_Chair_GoZA · 2024-12-21

**Metareview:**

This paper proposes an object-centric approach for RL. The idea is to assume that the abstract state consists of a set of items and their attribute. A strong assumption is made that in a transition, only one attribute can change or none although empirically the violations of this are tested. The agent explores using inverse-count reward bonus and MCTS is employed to explore.

Strengths:
- neat framework. I think object-centric representations make sense
- the proposed approach can transfer few-shot to new environments and object types

Weakness:
- strong assumptions on single-attribute change
- assumes access to an object-centric map that maps a visual observation to a set of encodings and a set of object-perturbing policies.
- assumes attribute values are discrete
- experiments are on simple visual problems (1980s/90s games style) although I understand that planning in these domains is quite challenging.

Overall, the core idea makes sense but it isn't too novel. It is similar to factored MDP which is an old RL paradigm (e.g., Guestrin et al., 2003, https://arxiv.org/pdf/1106.1822), although none of the factored MDP work is cited. Further, I agree that the assumptions are indeed strong. Some of these can be relaxed, e.g., there are lots of work on using features rather than counts for exploration (see Henaff et al., 2022 uses elliptic bonus which is cited). For others, such as the assumption of object-centric map, it will be great to show experiments with pre-trained models. E.g., can approaches like "Segment anything" (https://arxiv.org/pdf/2304.02643) be useful here to extract objects?

However, reviewers are positive on the paper and I think the RL community can benefit from a resurgence of object-centric mapping. I also
like the few-shot transfer approach. This is a desired property with significant benefit as we see with LLMs. Therefore, despite the many shortcomings listed above, I am recommending a weak accept. But I think this paper can benefit from a strong revision. I urge authors to consider moving away from discrete attributes: consider using continuous attributes and an elliptic-bonus exploration approach like that in Henaff et al., 2022. Also, consider using a foundation model instead of training a supervised learning model to train object mapping. And finally, please cite works on factored MDP and how your approach differs from them.

**Additional Comments On Reviewer Discussion:**

The main reviewer concerns were:

1. Strong assumptions of object mapping and policies: Author's response was that one can get this from foundation models, however, in the paper they train a classifier to predict attributes using supervised learning data something that is a strong assumption.

2. Choice of discrete attributes: Author's response was that discrete attributes cover a big space. I think ths might be true in some places, e.g., symbolic tasks in a computer but I don't think this is true in real-world robotics tasks which would be the flagship application for this topic. Discretizing positions, for example, can lead to a big increase in the search space, whereas using features can help model this in a more controlled way.

3. Assumption that only one attribute can change at a time: This is not always true. E.g., moving a block in a jenga game can result in all the blocks crumbling. In fact, these dependencies between objects are widely studied in factored MDP3. Authors show that the approach is resistant to this in Appendix B.2.2.

I think these points do matter and reduce the degree to which I want to suggest accepting the paper. However, reviewers also found the paper easy to read, liked the comprehensiveness of the experiments, and the generalization ability. Overall, I think the community can learn from this paper despite its many flaws.

---

### Decision · Program_Chairs · 2025-01-22

Accept (Poster)